# mRNA, a Revolution in Biomedicine

**DOI:** 10.3390/pharmaceutics13122090

**Published:** 2021-12-05

**Authors:** Bruno Baptista, Rita Carapito, Nabila Laroui, Chantal Pichon, Fani Sousa

**Affiliations:** 1CICS-UBI—Health Sciences Research Centre, University of Beira Interior, Av. Infante D. Henrique, 6200-506 Covilhã, Portugal; bruno.baptista@ubi.pt (B.B.); ritacarapito@gmail.com (R.C.); 2Centre de Biophysique Moléculaire (CBM), UPR 4301 CNRS, University of Orléans, 45071 Orléans, France; nabila.laroui@cnrs-orleans.fr

**Keywords:** mRNA, protein replacement therapies, immunotherapy, gene editing

## Abstract

The perspective of using messenger RNA (mRNA) as a therapeutic molecule first faced some uncertainties due to concerns about its instability and the feasibility of large-scale production. Today, given technological advances and deeper biomolecular knowledge, these issues have started to be addressed and some strategies are being exploited to overcome the limitations. Thus, the potential of mRNA has become increasingly recognized for the development of new innovative therapeutics, envisioning its application in immunotherapy, regenerative medicine, vaccination, and gene editing. Nonetheless, to fully potentiate mRNA therapeutic application, its efficient production, stabilization and delivery into the target cells are required. In recent years, intensive research has been carried out in this field in order to bring new and effective solutions towards the stabilization and delivery of mRNA. Presently, the therapeutic potential of mRNA is undoubtedly recognized, which was greatly reinforced by the results achieved in the battle against the COVID-19 pandemic, but there are still some issues that need to be improved, which are critically discussed in this review.

## 1. Introduction

After intensive work unveiling the DNA structure, its function, and potential biomedical applications, scientists changed their target and focused on the study of ribonucleic acid (RNA) [1,2]. Structurally, RNA is composed of a single polynucleotide strand, but some short regions within the RNA molecule can be complementary and form secondary structures (hairpins, loops, among others) [2,3] (Table 1). The secondary structure is determined by the base sequence of the nucleotide chain, so that different RNA molecules can assume different structures. Moreover, as RNA structure determines its function, this diversity is also responsible for RNA engagement in multiple cellular roles [2,4,5].

Cells contain an enormous variety of RNA forms, generally categorized as coding RNAs, which include the messenger RNA (mRNA), and non-coding RNAs (ncRNA) [2,3]. Although only 2% of the RNA transcribed from the human genome will encode proteins, it plays a fundamental role, directly or indirectly, in numerous biological processes [3,6], as it will be discussed later. Recently, with the recognition of the biological value of mRNA, it started to be exploited in a therapeutic perspective [2], gaining now increased relevance as an innovative vaccine for pandemic situations.

## 2. mRNA Structure and Biogenesis

Several types of RNA play fundamental roles in gene expression, a process responsible for the use of information stored in the DNA sequence to produce effector molecules, such as RNAs and proteins, which are the basis of cell biology [2]. The central dogma of molecular biology states that DNA is transcribed into mRNA, which is later translated into proteins [1]. The mRNA is essentially composed of a complimentary copy of the DNA and serves as a model for the production of proteins, as it contains the information about the amino acids sequence that will compose the target protein [2,7]. In addition, each mRNA molecule presents non-coding or untranslated sequences, which will control the mechanisms of processing and reading of mRNA [2,8].

In bacteria, mRNA is transcribed directly from DNA, and the untranslated region at the 5′ end (5′ UTR) of the mRNA chain, called the Shine-Dalgarno sequence, helps the mRNA bind to the ribosomes [2]. On the other hand, in eukaryotic organisms, the process of gene expression is more complex, involving RNA Polymerase II (Pol II) co for messenger RNA synthesis [9,10]. In this case, a pre-mRNA (the primary transcript) is formed, which is later processed to produce the mature mRNA [4]. In the mature mRNA molecule, each amino acid is encoded by a set of three nucleotides, called a codon [2]. The translation process and the stability of mRNA are determined by specific regions of RNA, which are of high importance for the improved durability of this biomolecule in the cell [8]. Both prokaryotic and eukaryotic mRNAs contain three primary regions [2,11,12], which can be observed in Figure 1. The first, 5′ UTR is a sequence of nucleotides at the 5′ end of the mRNA that does not encode for amino acids [13]. During the processing of pre-mRNA, the addition of a guanine nucleotide with a methyl group (CH_3_) to the 5′ end (Cap-5′), increases mRNA stability and assists in the transport of mRNA to the cytoplasm and further binding of the mRNA to the ribosome [11,13,14]. This modification facilitates translation, nuclear exportation [8,15], intron removal [2,8] and decreases immunogenicity, benefiting some types of therapies, such as in target proteins replacement. Modifications to the Cap-5′ end, which prevents its removal, may also result in an improvement in mRNA stabilization and resistance to enzymatic degradation [8,11,14]. The second region of the mRNA is the protein coding region, which comprises the codons that determine the amino acid sequence. The protein coding region starts with a first codon which encodes for methionine and finishes with a stop codon [2]. The modification of rare codons in the protein coding sequences with frequent synonymous codons, called optimization of codons, can result in altered levels of expression [8,15] and immunogenicity [16,17]. Both coding and non-coding RNA can be subjected to specific nucleotides’ modification, coined as epi-transcriptomic changes in eukaryotic cells [18]. Those modifications play an important role as they can affect the translation, initiation, stability, localization, or function of RNAs. In 2005, Kariko and Weissmann [19] demonstrated that it is possible to reduce the immunogenicity through chemical modifications of nucleosides of mRNA [8,12,13]. Adding modified nucleosides highly improves mRNA translation and vaccine performance, as described below [8].

The third region, similarly to the 5′ UTR, is a non-coding nucleotide sequence occurring at the 3′ end of the mRNA (3′ UTR) [2,17]. The modification of the 3′ UTR consists in the formation of a polyadenylated “tail” (Poly (A)), in a process called polyadenylation [2,11,12] which is represented in Figure 1. The poly (A) tail promotes inhibition of deadenylation by nucleases, leading to increased stability and a higher translation efficiency [20,21,22]. In addition, this sequence can be advantageous and has been highly explored for the purification of mRNA. Both 5′ and 3′ UTR regions increase the stability of mRNA [2,11], increasing the half-life of mRNA in the cell, which may lead to greater production of proteins [2,13,15]. Moreover, both UTRs of mRNAs are responsible for recruiting RNA-binding proteins, as well as microRNAs (miRNAs), and can profoundly affect translation activity [8,13].

## 3. Therapeutic Application: mRNA vs. DNA

In the 1990s, intracellular delivery technology and DNA or mRNA production methods were sophisticated enough to support preclinical studies on gene therapy, DNA- or mRNA-based vaccines and cancer immunotherapy [1,8,23]. Both DNA and especially mRNA serve as a model for protein synthesis and, if used in a therapeutic way, allow the cell to produce missing or defective proteins in a given disease or deficiency [15]. Therapies using these nucleic acids are more advantageous than therapies in which an absent or defective protein is directly applied, as they are relatively faster and cheaper to develop, without major size restrictions. In addition, certain proteins, such as membrane proteins, may be difficult or impossible to be effectively produced in vitro. Another advantage of using nucleic acids as biotherapeutics is related to production issues, more specifically the post-translational modifications that are mandatory in proteins and extremely difficult to be guaranteed by the production of proteins in heterologous systems [8,15]. The challenges on post-translation modifications may lead to significant differences between proteins produced endogenously and exogenously (in vitro) and, consequently, cause increased immunogenicity [1,12]. Moreover, the delivery of protein drugs has its limitations, mainly due to their large size and some instability, which makes it more difficult to reach in vivo the concentrations that can induce a therapeutic effect, not to mention the high production costs associated. In this sense, delivering exogenous mRNA into cells seemed to be a very promising way of manipulating protein expression, once it does not integrate cells’ genome, and its natural degradation pathway ensures its temporary activity [1].

Until few years ago, DNA-based therapeutic approaches were proposed as the most appropriate in the treatment of inherited diseases, which were caused by missing or dysfunctional proteins. However, the therapeutic use of DNA requires not only fast delivery of DNA to the cytoplasm but also an efficient entry into the nucleus, since DNA needs to be transcribed into mRNA before the therapeutic protein molecule can be produced [13,15,24]. Furthermore, since most differentiated cells are post-mitotic cells, which do not undergo frequent cell division, nuclear delivery is a major obstacle to DNA-based therapy [15,25]. Conversely, as mRNA enters the cytoplasm, the translation process can start immediately [26], without its transfer into the nucleus being necessary [12,16].

In 1970, mRNA was delivered to cells for the first time through liposomes [1,15]. At that time, mRNA was considered unfeasible for use for gene therapy purposes, due to its instability and immunogenicity [12,13,17]. In addition, at the beginning, mRNA was not recognized as having a relevant role in therapy, as it could not lead to permanent gene therapy [1]. This ideology has only changed in recent years, with the attainment of deeper knowledge about mRNA itself, as well as the identification of stabilizing modifications and the development of improved nucleic acids delivery technologies [13,15,17]. The evolution in this field has lead to an improvement in mRNA translation and half-life, as well as improved transfection efficiency, resulting in a significant reduction in immunogenicity [15,23,24]. Thus, mRNA, as a transient transporter of genetic information capable of inducing proteins expression, started to be investigated to replace DNA or recombinant proteins for therapeutic purposes [8,15,23]. Indeed, it was found that mRNA has a higher flexibility and greater therapeutic utility than almost all other classes of known drugs, mainly because of its transient nature [1,12]. This elevated the interest in mRNA to revolutionize different health areas, such as in vaccination, cell reprogramming, protein replacement therapies, and in the treatment of genetic diseases [8,24,27] in a safer way compared to other gene therapy strategies [1,13]. Since then, mRNA has evolved from a fragile molecule to a therapeutic agent in clinical studies [15]. Overall, this biomolecule, compared to DNA vectors, increases the transfection efficiency of inactive cells [26] and does not present the risk of insertional mutagenesis in the genome [13,25,28], making mRNA a more attractive molecule in protein supplementation therapy [17,24]. Moreover, mRNA enables a faster protein production, allows a higher control over the expression levels, and does not contain additional exogenous sequences, such as antibiotic resistances or viruses-derived promoters, highlighting the safety and effectiveness of mRNA [12,25,29]. The degradation of mRNA can be determined by its structure, thus allowing improved, controllable and temporary pharmacokinetics [1,12,15]. Another key point is the possibility to produce mRNA with different strategies, namely by chemical synthesis or by in vitro transcription, despite the limitations associated with these methods. An alternative that is being exploited is in vivo production, based on recombinant technology. However, for large-scale microbial production, there are still many challenges to overcome in order to achieve a safe process that could be totally accepted by regulatory agencies [12]. The use of mRNA also faces limitations related to its instability under physiological conditions and strong immunogenicity. One of the strategies that can help to solve this problem is the use of a drug delivery system (DDS) capable of protecting mRNA against nucleases and preventing recognition by the immune system, in order to achieve the desired therapeutic results [17,25]. As already mentioned, mRNA has a transient therapeutic effect [13,17]. Although this transient nature of mRNA is beneficial in certain cases, this feature also turns out to be a limitation [30]. In genetic and/or metabolic diseases, in which prolonged treatment is required, repeated administration of this biomolecule is necessary to sustain the therapeutic levels of a protein [8,15,24,30], and this can be stressful for the patient. However, regardless of the treatment regimen (transient or prolonged), the main goal of mRNA therapy is to deliver it successfully to the cytoplasm, for an efficient expression of the target protein, with negligible toxicity and immunogenicity [15]. In recent years, the therapeutic use of mRNAs has shown considerable promise both for prophylaxis (for example, antiviral vaccines) and for the treatment of a wide range of diseases, including myocardial infarction, human immunodeficiency virus (HIV) infection and several types of cancer [1,27], as will be discussed later.

## 4. Production of mRNA

Since mRNA has been exposed in a positive way to therapeutic actions, the challenges for mRNA production started to be addressed in a more intensive way. The most commonly used methods are the chemical synthesis and enzymatic production by in vitro transcription (IVT) of a linearized plasmid DNA (pDNA) [12,13,31], or by a PCR model [12]. These enzymatic strategies for obtaining mRNA depend on the presence of a bacteriophage promoter, 3′ and 5′ UTR, a protein coding region or open reading frame (ORF) (located between the two regions) [12,28,30], and optionally a Cap-5′ [13] and poly [d (A/T)] sequence [11,12,30]. These sequences (Cap-5′ and poly [d (A/T)]) help to obtain more stable mRNA, more efficient translation and easier transcript purification, due to the poly [d (A/T)] sequence [11,12,13].

The production of mRNA by IVT begins with the addition of all necessary nucleotides (which can be modified or not) [13] and the bacteriophage RNA polymerase (T7, T3 or SP6), which recognizes the promoter in the DNA model [12,25,31]. In eukaryotic cells, mRNAs have a tail on the 3′ UTR of 50 to 250 adenosine residues, and during the synthesis of mRNA, the inclusion of a poly (A) tail is also intended, in order to have a similar structure to endogenous mRNA. It has been reported that gradually increasing the length of the poly (A) tail of the synthetic mRNA to 120 adenosine residues could also increase the protein expression levels [11,12], however, longer chains (>120 adenosine residues) did not elicit the same effect [12]. If the DNA template for the IVT does not contain a poly [d (A/T)] sequence or if it is short, the poly (A) chain can be added post-transcriptionally by the poly (A) polymerase [12,23]. Although this method allows the addition of a poly (A) tail, it is not consistent, as it promotes a variable length of the poly (A) tail and therefore, mRNA molecules that originate will have poly (A) tails with heterogeneous lengths [12]. Synthetic mRNAs with a known and homogeneous poly (A) tail length can be obtained using a pDNA model. Still, it is difficult to know the exact size of this sequence due to the difficulties of sequencing long repetitive chains [12]. Due to these difficulties, pDNA models for mRNA IVT are often cleaved after the poly [(A/T)] sequence by IIS-type restriction enzymes. In this way, cleavage within the recognition sequence is avoided and prevents the existence of nucleotides other than adenines, which could result in mRNAs with a masked poly (A) tail [12].

As mentioned, in addition to IVT from a pDNA, mRNA can also be obtained by a PCR model. This model is produced from the IVT pDNA, using a primer upstream of the bacteriophage promoter and a second primer, which contains a poly (T) sequence, allowing its connection to the poly (A) tail at the 3′ end of the pDNA insert [12]. Regarding the addition of Cap-5′, it can be incorporated in the transcript during IVT, including a dinucleotide m7GpppG as the structural homolog of endogenous Cap-5′ [23]. After IVT, and in view of an adequate therapeutic effect, the synthetic mRNA should be purified, thus seeking the removal of any impurities, among which, the pDNA template, RNA polymerase, unincorporated ribonucleotides [12] and aberrant or double-stranded transcripts (dsRNA), which may have occurred in IVT. These impurities, if not removed from the synthetic mRNA of interest, may lead to unwanted immune responses or induce adverse effects in the organism upon administration [12,23,31]. In addition, if the synthetic mRNA is capped enzymatically, it needs to be purified to remove excess of “Cap” enzymes and molecules [12,25]. Among the most common purification methods used is chromatography [23,31,32]. This has the ability to remove ever-larger by-products, including DNA, which can be degraded by DNases, facilitating purification, as well as abortive transcripts and dsRNA [12]. Thus, a rigorous purification of the mRNA obtained by IVT was recognized as being one of the critical steps for obtaining pure and stable mRNA [23,32].

Globally, these methods allow the preparation of mRNA in a relatively quick and safe manner [24], but their cost is potentially high [33], so it might be beneficial to develop studies for recombinant mRNA production, along with purification strategies in order to obtain a recombinant, pure, safe and cheaper mRNA, which is already being applied to other RNA types [34,35,36].

## 5. Challenges in mRNA Delivery

To achieve the goal of an effective mRNA-based therapy, meaning the reestablishment of the production of a missing or defective protein, it is necessary that mRNA enter the cells, leading to the target protein expression at therapeutic levels. Usually, DNA and mRNA induce protein expression, whereas small interfering RNAs (siRNAs), microRNAs, oligonucleotides or aptamers provide gene silencing [37].

Due to its hydrophilicity, high molecular weight and negative charge, mRNA has poor cellular uptake in its free form. The knowledge of the mRNA structure allowed the optimization of specific modifications in order to improve the stability of the molecule, increase the efficiency of translation, and reduce immunogenicity. However, it is still necessary to overcome several extra and intracellular barriers for the effective therapeutic use of RNA [38], as presented in Figure 2. Based on the requirements for therapeutic action, the structural components of the mRNA molecule can be modified [13,15,26] (Figure 1). Although mRNA holds great promise in therapy for numerous diseases, the main obstacle is the ability to specifically deliver it to target cells.

One of the limitations of the use of mRNA as therapeutics is its susceptibility to enzymatic degradation, particularly by ubiquitous RNases present in the blood and tissues. Indeed, the existence of these nucleases is a significant obstacle to the systemic application of non-encapsulated mRNA [13,15,17]. In addition to RNases, there are other barriers to be surpassed, such as the cell membrane. Naked mRNA does not have the necessary characteristics for its entry through the phospholipid bilayer, as this biomolecule has a negative charge and a large size and is hardly transported into the cell without its incorporation into a delivery system [15,17,29]. Moreover, in in vivo applications, intravenously administered naked mRNA is rapidly degraded and may even cause an immune response. A study by Islam and colleagues found that the half-life of naked mRNA was about 5 min after intravenous administration, with a significant decrease in serum levels to 10% after 5 min and to 1% after 1 h [39]. Thus, mRNA encapsulation by appropriate delivery systems is an essential requirement for this molecule to overcome those barriers. Adding specific ligands that can recognize receptors on cell membrane will allow the improvement of mRNA delivery into target cells and tissues, with the final aim of achieving an enhanced and specific transfection [17,23,26,30].

Different types of in vivo administration routes have been used for mRNA delivery [15]. As for any type of nucleic acids, mRNA delivery can be achieved using main approaches comprising (i) physical methods that temporarily disrupt the cell membrane; (ii) viral-based approaches that use a virus or viral components capacity for cell transfection, and (iii) non-viral nanosystems [26] represented in Figure 3.

Physical methods are simple and consist of employing a physical force to the membrane barrier to deliver genetic material. The therapeutic application of naked mRNA has mainly been studied ex vivo by physical methods, including electroporation and gene gun [38]. These methods have started to be tested to increase the efficiency of mRNA uptake in vivo, but an increased cell death and limited access to target cells/tissues have been pointed out as disadvantages. For example, RNA complexed with gold particles showed good expression in tissues using gene gun in mouse models, however, no efficiency data are available for larger animals or humans [40,41].

Viral vectors are genetically modified viruses containing genes of a virus partially or completely replaced by antigen-encoding RNA [42,43]. Different viruses, such as picornaviruses [44] and lentivirus [45] have been chosen for mRNA delivery, given their high transfection capacity. In 2014, Hamann and co-workers described foamy virus vectors that allowed an efficient and transient expression of various transgenes by packing and transferring non-viral RNAs (both in vitro and in vivo) [33]. However, the use of recombinant viruses entails inherent risks, such as potential risks of insertion into genome, difficulty in controlling gene expression, limitations in the size of the sequence of interest and the existence of strong immunological side effects. Moreover, viral vectors present high production costs as well as a low packaging capacity [46,47,48]. Due to the risks associated with viral vectors, there has been an increasing number of studies on non-viral delivery systems. These have already demonstrated their enormous potential for the delivery of various nucleic acids, such as pDNA, siRNA and miRNA [26] and explored for the delivery of mRNA [8]. Due to their biocompatible and diversified properties, in addition to their simple formulation with mRNA, with adequate and controlled release kinetics [26], non-viral vectors seem to be the most promising systems. Non-viral systems are mainly made up of synthetic or natural biocompatible components that form complexes with mRNA and vary in composition, physicochemical characteristics, shape, and size [43,49].

Different types of non-viral systems (polymeric, lipid and hybrid systems) have been developed to protect and improve the delivery of mRNA. [43]. One of the main advantages of polymeric systems is the possibility of their modification to adapt them to the active substance. Biodegradable and mainly cationic polymers are commonly used to develop nanoparticle formulations with nucleic acids, leading to the formation of polyplexes [42,43]. The gold standard of polymeric vehicle, Polyethylenimine (PEI) has been proposed for mRNA delivery but its use has been limited due to its toxicity [50]. New PEI derivatives were designed to improve their biocompatibility and transfection efficiency. For example, a study by Forrest and colleagues described that the toxicity of the PEI polymer is reduced by its conjugation with cyclodextrin [51]. Another example is jetPEI^®^, a linear PEI marketed as an in vivo mouse transfection reagent designed to improve its biocompatibility and transfection efficiency. This derivative was recently evaluated by Sultana research group, in the administration of mRNA by direct myocardial injection in mice, demonstrating the expression of the protein in the lungs [52]. Moreover, in 2016, Démoulins and co-workers incorporated mRNA encoding influenza virus hemagglutinin and nucleocapsid into PEI nanoparticles in order to improve mRNA vaccines efficacy which was confirmed by both humoral and cellular immune responses induction, in vivo [53]. In addition to PEI, there are other polymers under study. Zhao and colleagues developed a polyethyleneimine-stearic acid (PSA) copolymer to deliver mRNA encoding HIV-1gag into dendritic cells and BALB/c mice. With this polymeric system, and after its subcutaneous injection, Zhao group was able to detect specific antibody levels for anti-HIV-1gag [54]. Another example of a promising polymer is chitosan, a biocompatible cationic glycopolymer that showed ideal conditions for delivering mRNA encoding luciferase [55]. McCullough and colleagues also investigated whether chitosan was able to deliver the self-replicating replicon-RNA encoding the influenza virus hemagglutinin [56]. However, when biodegradable polymers, such as poly Lactic-co-Glycolic Acid (PLGA), are negatively charged, they cannot effectively encapsulate negatively charged mRNAs. Even so, this polymer, when hybridized with cationic lipids, already allows the encapsulation of mRNA [39].

Although widely studied for mRNA complexation, cationic polymers are not as clinically advanced as lipid systems for mRNA-based therapies [42,43]. Various synthetic and naturally derived lipids are used in mRNA delivery. Liposomes have an inherent advantage, as they are able to mimic the composition of the cell membrane, but they are not extensively used as the majority of lipoplex formulations cannot stand the presence of serum and mRNA is more unstable [42,57].

Currently, lipid nanoparticles (LNPs) are considered one of the most developed systems for mRNA delivery and are at the forefront of clinical trials [43]. Typically, LNPs are made up of an ionizable lipid, neutral/auxiliary lipid, cholesterol and PEG lipid. Ionizable lipids are lipids that become cationic at acid pH, allowing them to interact with mRNA through electrostatic interactions, leading to the formation of a complex called lipoplex, while at physiological pH, they have a neutral charge, highly reducing their toxicity [42,43]. Patisiran (ONPATTRO™), the clinically approved siRNA formulation against transthyretin mRNA is made of LNPs containing Dlin-MC3-DMA (MC3 ionizable lipid) [58]. Within LNPs, in addition to ionizable lipids, several studies demonstrate the use of cationic lipids for mRNA encapsulation [59]. N-[1-(2,3-dioleyloxy)propyl]-N,N,N-trimethylammonium chloride (DOTMA) was the first synthetic cationic lipid used to complex in vitro transcribed (IVT) mRNA [60]. In addition to this, one of its derivatives, 1,2-dioleoyl-3-trimethylammonium-propane (DOTAP) has also been studied for mRNA delivery [61,62]. However, cationic lipids tend to reduce efficacy due to possible nonspecific interactions in vivo, in addition to possible inflammatory responses and toxicity [63] Initially, LNPs were considered promising siRNA delivery systems [64]. Several groups used this same ionizable lipid for LNP dedicated for mRNA delivery [65,66]. Pardi and co-workers report that a single intradermal immunization of LNP encapsulating nucleoside-modified mRNA encoding for the pre-membrane and envelope glycoproteins of a strain of Zika virus elicited potent and durable immune responses in mice and in non-human primates [67]. Moreover, Hekele et al. showed that mRNA encoding the HA antigen of the influenza virus H1N1, encapsulated in LNP made of 1,2-dilinoleyloxy-3-dimethylaminopropane, 1,2-diastearoyl-sn-glycero-3-phosphocholine, cholesterol and PEG-DMG 2000, rapidly induced immune responses in mice [68].

LNPs are now considered as the gold standard of mRNA formulation as they are currently used to prepare COVID-19 vaccines from both Moderna and Pfizer/BioNTech (discussed in Section 6.2).

In order to take advantage of the various delivery systems, the hybrid system, that is, the combination of different materials, such as polymers, lipids, among others, offers greater functionality and flexibility than isolated systems. Lipopolyplexes are an example of such a hybrid system in which the advantages of cationic polymers and lipids for the complexation of nucleic acids are combined [43,69]. The first example of this hybrid system is made of polyethylene glycol (PEG)ylated derivative of histidylated polylysine and L-histidine-(N,N-di-n-hexadecylamine)ethylamide liposomes reported by Mockey and collaborators [70] demonstrated that intravenous injection of MART1 mRNA encapsulated by this lipopolyplex induced a specific vaccination of mice with significant protection against B16F10 melanoma tumor progression. In addition to these most used delivery systems, others have also been explored, such as polypeptides [71,72,73,74], and mineral-coated microparticles [75]. CureVac has developed the RNActive^®^ technology which makes use of protamine as an mRNA delivery system. This is an mRNA vaccine platform based on protamine/mRNA complexes, which is currently under clinical evaluation [72,73,74], and at pre-clinic stage against influenza virus [71]. In 2016, in phase I clinical trials, Kranz and co-workers showed that RNA-lipoplexes, without functionalization of the particles with specific ligands, could precisely target dendritic cells, just by adjusting its net charge. This particular study can be the basis of the development of a class of vaccines for dendritic cells targeting cancer immunotherapy [76]. In 2020, Mai and co-workers developed a cationic liposome/protamine complex (LPC) for nasal administration of mRNA vaccines. In vitro, LPC containing RNA encoding cytokeratin 19 presented significantly greater efficiency in the uptake of vaccine particles and provoked a strong immune response and slowed tumor growth in an aggressive lung cancer model [77].

Regardless of the type of transfection, in vitro (ex vivo) or in vivo, mRNA is translated directly into a functional form, promoting a faster and easier response and therapeutic application [17].

## 6. mRNA Therapies

Currently, mRNA appears as a very promising and innovative therapeutic approach for diseases associated with functional loss of proteins, through the administration of a synthetic mRNA, which promotes the reestablishment of protein levels and restores its function [17]. Moreover, mRNA can create new cellular functions, for example for passive immunization [13,23], allowing to stimulate the immune system, through the translation of antigenic mRNA for specific cell recognition (e.g., cancer cells) or antibody production [15,24]. The fact that a relatively small amount of encoded antigen, from a synthetic mRNA, can be sufficient to obtain robust signs of efficacy, is one of the main advantages of using this biomolecule in immunotherapy [26]. However, the global success of such mRNA-based treatments depends on a high number of these biomolecules and an effective in vivo delivery to target cells involved in a given disease [13,23,30]. After proving that in vivo mRNA administration is possible and viable, the concept of using mRNA as a therapeutic basis was readily accepted and used [11,13] in a variety of diseases, including diabetes, HIV infection, anemia, hemophilia, myocardial infarction, cancer, asthma, metabolic disorders, fibrosis, skeletal degeneration and neurological disorders, such as Friedreich’s ataxia and Alzheimer’s disease [1,11,17].

Initial mRNA therapies centered its use against cancer, transferring this biomolecule ex vivo and then promoting a localized delivery in vivo (Figure 4). Currently, the most advanced therapies using mRNA are in the clinical testing phase (see Section 6.2). Unlike cancer vaccines and immunotherapies, most protein replacement therapies that use mRNA are still in preclinical development [15].

### 6.1. Protein Replacement Therapies

Protein replacement therapies performed by mRNAs have enormous potential for treating a wide range of diseases [1,8,17,29] (Table 2).

Application of IVT mRNA for protein replacement therapies relies on the supplementation of proteins that are under-expressed or not functional, as well as on the expression of foreign proteins that can either activate or inhibit certain cellular pathways. Therapies based on mRNA have become more attractive because, contrarily to DNA, mRNA does not enter the nucleus of host cells, and therefore does not present a risk of genome integration or mutagenesis. There are different applications of this type of therapy including genetic and rare diseases [43,78,79]. mRNAs are generally designed to express therapeutic proteins, in such a way as to exhibit no or low immunogenicity, to have prolonged stability and high translation efficiency [8]. Most mRNA-based protein replacement therapies are targeted at certain organs, such as liver, lungs and heart, mainly because currently existing methods are more efficient for the delivery of mRNA to these tissues. The use of this therapy in other organs and cell types requires the development of new delivery strategies, active targeting or different methods of administration [8,29,30].

An example of protein replacement using IVT mRNA is a study performed by Baba and co-workers. In this work, mRNA has shown to be promising in treating neurological disorders by providing proteins and peptides in their native and mature form in neural cells. By using novel mRNA-loaded nanocarriers and carrying out administration through nasal route into mouse models, there was a sustained protein expression for almost two days in nasal tissues. Moreover, upon daily intranasal administration, neurological recovery of olfactory function was enhanced, as well as recovering almost to a nearly normal structure of the olfactory epithelium [80]. In 2017, Ramaswamy and co-workers successfully delivered mRNA through LNP in order to treat a Factor IX (FIX)-deficient mouse as a model of hemophilia B. This study showed that repeated administration of the mRNA-LNPs complex did not cause innate immune responses in hemophilic mice [78]. In 2018, Magadum’s team verified that modified mRNA (modRNA) could induce cardiomyocytes (CMs) proliferation and regeneration by upregulating mutated human follistatin-like protein 1 (hFSTL_1_). The post-translational modification was hypothesized to be responsible for CM regeneration in vitro with no indications of cardiac hypertrophy. Furthermore, it significantly improved cardiac function, decreased scar size, and also increased capillary density, showing the effectiveness of modRNA in CM proliferation and cardiac regeneration [81]. The same authors, in 2020, also showed that modRNA can induce CM cell cycle by upregulating the glycolytic enzyme pyruvate kinase muscle isoenzyme 2 (Pkm2). This increased expression of the enzyme contributed to re-enforcing the CM cell cycle, which led to cell division and consequently cardiac regeneration [82].

### 6.2. Immunotherapy

In addition to protein replacement therapy, another branch is immune stimulation against certain diseases. Interleukin 15 (IL-15) cytokine presents a therapeutic anticancer potential, mainly for its immunologic stimulation properties. However, currently used delivery systems with pDNA present low efficiency, and the use of in vitro transcripts could be a better solution. For this, Lei and co-workers, in 2020, verified that through cytokine expression with this mRNA, lymphocyte stimulation was successfully produced and cytotoxicity was triggered in cancer cells. Local or systemic administration of this mRNA induced inhibition of cellular proliferation in several colon cancer models in a safe and efficient way. These results have shown the high therapeutic potential for colorectal cancer immunogenic therapy with this approach [83]. In the same context, Interleukin 2 (IL-2) exerts significant anti-tumor activity. This cytokine is involved in proliferation, differentiation and effector function of T cells and since 1998, it has been approved for the treatment of metastatic melanoma [84]. However, using IL-2 cytokine faces several limitations including the short serum half-life. For this, the use of mRNA expressing IL-2 would prolong the production of the cytokine, thus reducing high and frequent doses. Currently, two nucleoside-modified mRNA LNP encoding for IL-2 are in clinical development for cancer treatment (see review [85]).

Vaccines have been used to provide adequate, specific and short-term immune responses against infectious diseases or cancer. Conventional vaccines consisting of attenuated microorganisms or that contain the majority of virus or bacterial antigens have demonstrated lasting protection against a variety of infectious pathogens, but on rare occasions they can revert to their pathogenic forms [1,8]. More and more epidemic outbreaks are caused by viral infections and in all cases, those are characterized by their unpredictability, high morbidity, exponential spread, and substantial social and economic impact [86]. mRNA vaccines, on the other hand, cannot replicate within the body [1]. Thus, mRNA vaccines have been deeply investigated due to their ability to encode a wide range of antigens, due to the self-adjuvant effects [8,23], as well as for their potential large-scale production in a fast, flexible and low-cost manner [8,23,86]. The development of an mRNA vaccine for specific antigen immunity requires the transfection of antigen-presenting cells, such as dendritic cells [1,8,87], resulting in the induction of humoral and cytotoxic T-cell response [86] which is represented in Figure 5. Because of this, the administration is typically performed by intradermal, intramuscular or subcutaneous injection, as dendritic cells are densely found in skeletal muscle and skin tissue [1,8,29]. In addition to mRNA, DNA was also used for the coding of antigens, but due to the potential for integration into the genome, its use was rather limited [8,23].

In contrast to mRNA, antibody-based cancer therapy faces some challenges related to antibody production problems, low stability in long-term storage, aggregation, and the presence of several impurities intrinsic to the production process. In addition, antibodies, especially bispecific antibodies, have a low serum half-life and continuous administration is required to achieve the therapeutic effect [23]. Thus, the use of mRNA for the generation of therapeutic antibodies in patients represents a promising approach, in order to overcome the limitations of direct use of recombinant antibodies [23].

The development of an mRNA vaccine consists of acquiring genetic information of the infectious agent or the sequence of antigens associated with the tumor. Then, the gene is sequenced, synthesized and cloned into a plasmid. The mRNA is transcribed in vitro and the vaccine is administered to the patient [86]. The mRNA vaccine uses the host cell machinery to translate the corresponding antigen mRNA sequence, thus mimicking the infection or a tumor cell, in order to elicit humoral and cytotoxic immune responses [86,88]. The use of mRNA to induce adaptive immune responses in cancer (examples of vaccines presented in Section 6.2) began in 1995, with the discovery of protective antitumor immunity, which was obtained by intramuscular injection of mRNA from the carcinoembryonic antigen [89]. There are two main types of mRNA immunotherapy against cancer. The first type of immunotherapy works at the cellular level, in the same way as an mRNA vaccine, however, the mRNA encodes tumor-associated antigens. The second type, on the other hand, involves the modification of T cells, with chimeric antigen receptors (CARs), which is called CAR T cell therapy. Billingsley and collaborators demonstrated the C14-4 LNP induced CAR expression at levels equivalent to electroporation, with a substantially reduced cytotoxicity. When compared to electroporated CAR T cells by the lipid system, C14-4 LNP, in a coculture assay with Nalm-6 acute lymphoblastic leukemia cells, Billingsley and collaborators found that both methods induced a strong cancer-killing activity [90]. These results obtained by Billingsley research group show the progress that has been mad and the promising strategies to deliver mRNA to T cells. Usually, in this class of immunotherapy, the patient’s T cells are transfected with synthetic mRNAs, encoding CARs, bind to specific tumor antigens, subsequently eliminating the tumor cells [1].

Since then, mRNA vaccines have been classified into two subtypes: (i) non-amplifying mRNA-based vaccines (also known as mRNA conventional vaccines), that encode the antigen of interest and contain the 5′ and 3′ UTRs [88]; and (ii) self-amplifying mRNA (SAM or saRNA) vaccines [8,13,86,91] that not only encode the antigen, but also the viral replication mechanism, allowing an increase in the amount of intracellular mRNA, consequently leading to a more abundant protein expression [88] (Table 3 and Figure 6). Both types of mRNA vaccines use the translation mechanism of host cells to produce target antigens, in order to induce specific adaptive immune responses [8]. In 2020, He and co-workers developed cationic nanolipoprotein particles (NLPs) to enhance the delivery of large self-amplifying mRNAs (replicons) in vivo. These cationic lipids successfully encapsulated RNA encoding luciferase, protected it from RNase degradation and promoted replicon expression in vivo [92].

The greatest barrier to the usefulness of these vaccines is the need for intracellular delivery [8,13]. However, as already mentioned, through chemical modifications, encapsulation by nanoparticle formulations and through sequence engineering, it is possible to promote an improved targeting, delivery and entrance into the cell, in addition to greater efficiency in translation and enhanced half-life of synthetic mRNA vaccines [8]. Chronologically, mRNA vaccines in dendritic cells for adaptive immunotherapy against cancer and protein replacement therapies were the first therapeutic applications with these biomolecules to enter clinical trials [8,13]. Although therapies based on dendritic cells still represent the majority of clinical trials of mRNA vaccines, vaccination with the use of this biomolecule through non-viral vectors, as well as gene editing, is increasingly being investigated in search of new therapies against diverse diseases [1,8] (Table 4).

The most recent case of immunotherapy associated to mRNA vaccination, in clinical trials, was registered in 2020 and concerns the virus named “Serious Acute Respiratory Syndrome Coronavirus 2 (SARS-CoV-2)”, the former being known as Coronavirus (COVID-19 or 2019-nCoV) [112,113]. SARS-CoV-2 causes an infection in the alveolar epithelial cells of the human respiratory tract [114,115]. SARS-CoV-2 has a large genetic structure. The genome is surrounded by helical nucleocapsid proteins (N) and an outer envelope composed of matrix or membrane glycoproteins (M), envelope proteins (E) and spike glycoproteins (S) (Figure 7), which improve binding to cells, transport and interfere with the immune response of the host [91,116,117]. In addition, the virus has several non-structural proteins (NsPs) that are vital for its life cycle and pathogenic character [91].

The S glycoprotein is part of the outer layer of the virus and is essential for its entry into cells [118]. This protein consists of a receptor binding domain (RBD), that is responsible for specific binding to the angiotensin-converting enzyme 2 (ACE2) receptor, thus allowing the entry of SARS-CoV-2 [114] in the epithelium cells of human lung [119]. In addition, there are studies that indicate that SARS-CoV-2, as well as SARS-CoV, can enter the cell through clathrin-mediated endocytosis [115]. Of all structural proteins, it was found that S glycoprotein induces neutralizing antibodies and it was the main target antigen for the development of the vaccines [117,120].

Given the high transmission of SARS-CoV-2, the World Health Organization (WHO) emphasized the demand for a rapid response to this situation, endeavoring the immediate development of safe and effective prophylactic therapies [112,121]. Due to the great technological advances in sequencing techniques, it was possible to obtain colossal knowledge about SARS-CoV-2 in a very short time, something unprecedented in the history of medicine [113,118].

Vaccines decrease the viral spread and transmission from person to person [88,91], and the development of the SARS-CoV-2 mRNA vaccine was impressively fast [122]. The classical development of vaccines requires an average of about 5 to 10 years, but given the need and technological advances, the development time of the vaccine against SARS-CoV-2 was substantially shorter [88,113].

The approved mRNA vaccines to combat SARS-CoV-2, the vaccines developed by Moderna/NIAID and BioNTech/Fosun Pharma/Pfizer, aim at the expression of the S glycoprotein or RBD subunit [88,91,123]. After transfection of either muscle cells or dendritic cells, the expressed S glycoproteins are presented by the major histocompatibility complex (MHC) class I and II [113]. This process stimulates humoral immunity and leads to the production of neutralizing antibodies against the S glycoprotein by B lymphocytes, preventing viral binding and entry into cells [122] represented in Figure 8. It also induces the generation of specific cytotoxic T cells (CD8^+^) which can eradicate SARS-CoV-2-infected cells [124,125].

To date, the Pfizer/BioNTech (Comirnaty) vaccine presents 95% of efficiency, while the Moderna (Spikevax) vaccine has 94.5%, Gamaleya product has 92% and the AstraZeneca vaccine, 70% [126,127]. The first two (Pfizer/BioNTech and Moderna) are RNA vaccines that express COVID-19 spike glycoprotein, while the Gamaleya and AstraZeneca vaccines express spike protein from adenovirus vector platforms [128,129].

Both Moderna and Pfizer/BioNTech vaccines are made of a nucleoside-modified (N1 methyl pseudouridine) mRNA formulated in LNPs. These LNPs contain an ionizable lipid, neutral/auxiliary lipid ((phospholipid distearoylphosphatidylcholine) (DSPC)) at physiological pH and cholesterol, which allows to stabilize LNP and increase the efficiency of mRNA delivery, and finally a polyethylene glycol (PEG), which aims to improve colloidal stability, reducing opsonization by plasma proteins. However, these differ in the use of the ionizable lipid, as the Pfizer/BioNTech vaccine used the ionizable lipid ALC-0315 and Moderna vaccine used another ionizable lipid, the SM-102 (Table 5). Although both vaccines use ionizable lipids, both are tertiary amines that are protonated at a low pH, thus allowing for mRNA interaction and protection [130,131] (Table 5). These specific LNPs are therefore essential for a safe and efficient immune response. The mRNA encodes the membrane-anchored, full-length SARS-CoV-19 spike protein and contains mutations for the prefusion conformation, which stabilize the Spike protein.

The LNPs prevent RNA degradation and enable its delivery into host cells after intramuscular injection. Once inside the host cells, mRNA is translated into SARS-CoV-2 spike protein. The expression of this spike antigen induces neutralizing antibodies, as well as cellular immune responses against it, which can confer protection against COVID-19 [132]. The Pfizer/BioNTech vaccine has been recommended to people older than 12 years old, with a dose of 30 μg (0.3 mL) at a cost of $19.50 in the US. Recently, FDA issued emergency use authorization in individuals 5 years of age and older [133]. It consists of a two-dose administration with 21 days between each administration, providing immunogenicity for at least 119 days after the first vaccination and is 95% effective at preventing the SARS-CoV-2 infection. However, the Moderna Vaccine (mRNA-1273) has been recommended to people of or above 18 years of age, with a dose of 50 μg (0.5 mL) at a cost ranging from $32 and $37, in the US [134]. Similarly, it also consists of two shots administered 28 days apart providing immunogenicity for at least 119 days after the first vaccination and is, as referred above, 94.5% effective in the prevention of SARS-CoV-2 infection. It should be noted that age-dependent administration is region-specific, and can vary in different countries [135,136]. Given this, it is safe to say that both vaccines are beneficial in providing immunity against SARS-CoV-2 infection, nevertheless, some allergic responses have been reported. These COVID-19 vaccines can cause mild adverse effects after the first or second dose, including pain, redness, swelling or itching (at the site of vaccine injection), fever, fatigue, headache, muscle pain, nausea, and rarely cause anaphylactic shock. The Pfizer/BioNTech vaccine reports a lower percentage of these adverse effects comparatively with the Moderna vaccine. However, the Moderna vaccine is easier to transport and store (storage between −25 °C and −15 °C) because it is less sensitive when compared to the Pfizer vaccine (stored between −80 °C and −60 °C) [127,137,138].

CVnCoV (CureVac) consists of LNP-encapsulated non-chemically modified mRNA with naturally occurring nucleotides encoding for a full-length S protein that includes two proline mutations (S-2P), which was previously showed to stabilize the conformation of the S proteins for SARS-CoV. The mRNA was codon-optimized in order to provide a higher expression level of S protein and a moderate activation of the immune system [139]. The CureVac vaccine can be distinguished from the previous two candidates by exclusively consisting of non-chemically modified nucleotides and can be applied at comparatively lower doses (12 μg). CureVac company announced preliminary data on 16 June (from a 40,000—person trial), that its two-dose vaccine was only 47% effective at preventing COVID-19, which is half of the efficiency of its previous two rivals. It was expected that this third vaccine candidate would be cheaper and last longer in refrigerated storage than the earlier mRNA vaccines made by Pfizer/BioNTech and Moderna. However, it is suspected that CureVac’s decision not to exchange the biochemical structure of its mRNA, as Pfizer/BioNTech and Moderna did, might be the reason for its poor performance [139,140].

It should be noted that clinical application of mRNA as a therapeutic agent has some limitations due to its instability and the capacity to activate the immune system. Therefore, modifying the in vitro transcribed mRNA structure alongside with the design of suitable nanoparticles is of great importance [43]. This fact comes to be noticed because Pfizer/BioNTech and Moderna vaccines call upon modified RNA, by replacing uridine itself for another nucleotide called pseudouridine (Ψ), which is similar to uridine but contains a natural modification. This modification in exogenous mRNA is thought to decrease inflammatory reactions, while improving translational efficiency and stability. In contrast to Pfizer/BioNTech and Moderna vaccines, CureVac uses normal uridine instead of Ψ, which could be a reason for its poor success once higher doses reflected more severe adverse effects [19,141,142,143]. These improved properties conferred by the incorporation of Ψ make mRNA a promising tool for both gene replacement and vaccination. The innate immune system cells are activated by RNA, since it stimulates Toll-like receptors (TLRs), namely TLR3, TLR7, and TLR8. However, when some modified nucleosides, like, Ψ, 5-methylcytidine (m5C), N6-methyladenosine (m6A), 5-methyluridine (m5U), or 2-thiouridine (s2U) were included into the transcript, most of TLRs were no longer activated, therefore controlling immune activation in vitro and in vivo [19]. These characteristics and the readiness of producing such RNAs by in vitro transcription make Ψ-containing mRNA an important tool for the expression of any protein [144,145,146,147]. Furthermore, codon optimization strategies have been investigated to improve the cost efficiency of recombinant protein production, once most amino acids are encoded by different codons. This is primarily based on the substitution of multiple rare codons by others more frequent, that encode the same amino acid, thus resulting in increased rate and efficiency of protein translation [148]. Another successful modification is the addition of a poly(A) to IVT produced mRNA, which can be directly added during the transcription process (if the DNA template encodes the poly(T) sequence) or can be added post-transcriptionally by enzymatic reactions. Poly(A) tail length influences stability and translation efficiency. Even with a relatively long poly(A) tail that seems to be appropriate, the optimal length can vary depending on the target cell [43,149]. The Kozak sequence plays a major role in the initiation of the translation process and is located at the 5′ UTR. This sequence, defined as “RCCAUGG”, where “R” stands for a purine (A or G), helps drive high levels of translation from the correct start codon, therefore being considered the election sequence for translation initiation in eukaryotes [43,147]. The Pfizer/BioNTech vaccine included a poly(A) chain in their mRNA sequence, as well as an optimized Kozak sequence [147].

Conventional vaccine approaches, such as the use of attenuated and inactivated viruses, successfully provide durable protection against infectious diseases, but they are not able to meet the need for rapid and large-scale development. As already mentioned, although genetic immunization, such as DNA vaccines, has shown to be promising, pDNA delivery raises safety concerns due to the possibility of insertional mutagenesis. Thus, in order to try to obtain a vaccine quickly, safely and effectively, the development of an mRNA vaccine seems to be a reliable approach. This is a safer alternative, as it does not require entry into the nucleus for translation to occur, leading to an improvement in transfection and expression efficiency compared to DNA vaccines. It also presents comparatively lower production costs and capacity for rapid development, because with a simple change of the mRNA sequence, it will lead to the expression of a different protein, which is beneficial given the frequent viral mutations [91,112]. There are currently eight mRNA-based vaccines in clinical development and 22 in pre-clinical studies (Table 6) [150].

### 6.3. Gene Editing

As previously mentioned, mRNA therapies may also function in gene editing, which can be achieved by encoding nucleases from mRNA for cellular reprogramming [1,13]. Gene editing involves the precision of “cutting” and “pasting” genomic DNA in specific locations, expecting the establishment of a potentially permanent cure for genetic diseases to be a promising therapeutic area for the application of mRNA technology [1,8]. In this therapeutic area, mRNA function is to express programmable nucleases, including zinc finger nucleases (ZFNs), transcription activator effector nucleases (TALENs) [8,13] or CRISPR-Cas9 [1,8]. These genetic engineering tools allow the replacement or modification of gene expression, through the introduction or local deletion of specific modifications in the genome of target cells [8]. This allows the correction of a target gene, by deleting disease-causing mutations or by inserting protective mutations by joining the non-homologous end (NHEJ) [151,152] or even performing a repair or insertion directed to homology (HDR) [151,152,153]. This is schemed in Figure 9. ZFNs and TALENs facilitate the recognition of a sequence by protein–DNA interactions [8,13], however, the complex engineering necessary to create specific domains in proteins directed to DNA recognition and binding greatly restricts its wide application. However, the CRISPR-Cas9 system in Figure 9, is currently the most widely used and characterized gene editing technology [1,8].

In 2020, Jennifer Doudna and Emmanuelle Charpentier won the 2020 Nobel Chemistry Prize for their discovery of a novel and innovative gene-editing technique. CRISPR-Cas9 gene-editing tools allow precise editing of the genome and have countless applications, which scientists aim to use to alter human genes to eliminate diseases and eradicate pathogens. CRISPR-Cas9-mediated gene editing requires only two components: Cas9, a nuclease responsible for DNA cleavage and a short single-stranded RNA guide (sgRNA), which directs DNA cleavage by the nuclease, precisely. Typically, these two components are delivered to cells using a pDNA containing the Cas9 protein and sgRNA genes [1,153]. For more information on the mechanism of action of the CRISPR-Cas9 system, the following literature can be analyzed [151,152,153]. The use of CRISPR-Cas9 technology had only been used to edit the genomes of embryos, zygotes, and cultured cells [154,155], however, this technology has been increasingly used in vivo.

Due to the transient nature of mRNA, the use of this biomolecule can be advantageous in relation to the use of pDNA [1], limiting the presence of nucleases inside cells [13]. In this way, there is a reduction in possible non-specific cleavages which decreases the immune response to the Cas9 protein. In addition to these advantages, it appears that the intracellular presence of the Cas9 protein has been more persistent after mRNA expression compared to the administration of the Cas9/sgRNA ribonucleoprotein complex (Cas9-RNP) [1,8]. As such, co-delivery of mRNA, which encodes Cas9, and sgRNA is an attractive alternative [1]. Cas9 can be administered as mRNA, plasmid DNA or even as a protein. However, for plasmid DNA Cas9 to be functional it must overcome cell and nuclear membrane barriers. Thus, an alternative approach could be the use of Cas9 mRNA. This approach becomes preferable as mRNA only needs to cross the cell membrane to be functional. Liang and collaborators, using the GeneArt commercial system (Thermo) and electroporation, found that in the study of eleven cell lines, the delivery of Cas9 mRNA/gRNA or Cas9 RNPs was superior to the plasmid delivery in all cell lines tested. They also noticed that although the similar cleavage kinetics between Cas9 delivered as plasmid DNA, mRNA and protein to HEK293 cells, in cells transfected with plasmid DNA, the Cas9 protein accumulated over time, while the relatively low expression of Cas9 in mRNA-transfected cells seemed relatively stable for approximately 48 h. However, due to the fast turnover of Cas9 RNP and mRNA compared to the long persistence of Cas9 expressed from plasmids, this could reduce the opportunity for off-target binding and cleavage. When studying this among the six potential off-target sites, it was observed that the use of mRNA and Cas9 RNP had much smaller off-target effects than the use of Cas9 from plasmid DNA [156]. In addition to the use of mRNA in gene editing for the treatment of acquired diseases, Mohsin and colleagues demonstrated by in vitro experiments that Cas9 mRNA/sgRNAs can reduce the sporulation percentage of *E. tenella* oocysts, as well as their survival rate. These data show that the use of a highly specific sgRNA molecule, when combined with Cas9 mRNA, may be a potentially powerful agent in the development of new therapeutic drugs against parasitic diseases [157]. It should be reinforced that for long-term gene therapy purposes, mRNAs are not sufficiently stable; nevertheless, even transient articulation and ability will make hereditary change perpetual for the activity of Cas9 nuclease. This is the reason why Cas9 mRNA is commonly used, for example in Drosophila, zebrafish, Xenopus and mouse, in both cell culture and model organisms [158,159,160,161]. In addition to the use of non-viral systems, Ling and colleagues found effective and successful delivery using a viral system. These authors used mLP-CRISPR, a lentiviral system that delivers mRNA encoding one of the longest Cas9 proteins (SpCas9) and gRNA simultaneously. By targeting vascular endothelial growth factor A (Vegfa), it was found that with only a single sub injection-retinal of mLP-CRISPR in mouse models, 44% of Vegfa in retinal pigment epithelium was knocked out and the area of choroidal neovascularization was reduced by 63% without inducing off-target edits or anti-Cas9 immune responses [45]. Although CRISPR-Cas9 is most used to control over-expression levels of a particular protein, Qiu and colleagues verified the knockdown of the Angiopoietin-like 3 (Angptl3) gene in a specific and efficient way using the system CRISPR-mRNA Cas9, which led to a significant reduction in serum Angptl3 protein, low-density lipoprotein cholesterol (LDL-C), and triglycerides (TG) levels in wild-type C57BL/6 mice [162], presenting similar results to studies with antisense oligonucleotides [163]. They also verified that the therapeutic effect of genome editing was stable for at least 100 days after the single dose administration [162]. Wang and collaborators demonstrated that CRISPR/Cas9 mRNA-mediated gene editing technology allowed the simultaneous disruption of five genes in mouse embryonic stem cells (ES) with high efficiency, thus verifying that with Cas9 mRNA co-injection and sgRNAs targeting Tet1 and Tet2 in zygotes achieved mutations in both genes with an efficiency of 80% in mice with biallelic mutations [159]. This discovery not only allows us to verify that the CRISPR/Cas9 technology allows mutations in several genes simultaneously, but it will also greatly accelerate the in vivo study of functionally redundant genes and epistatic gene interactions.

Since the discovery of CRISPR-Cas9, the CRISPR revolution has expanded beyond its original use as a genetic engineering tool. New Cas nucleases are being developed to enable faster and more accurate molecular diagnostic platforms for use with next-generation bio-sensing platforms. Abbott and co-workers showed, through bioinformatic analysis, that some different CRISPR-associated RNAs (crRNAs) were able to target over 92% of live influenza A virus strains and over 91% of all coronaviruses. This fact expands CRISPR-Cas13 systems applications beyond diagnostics, such as SHERLOCK, and live-cell RNA imaging [164]. In this context, Cas 13 is an endonuclease that targets and binds sg RNA. Moreover, it demonstrates RNA cis-cleavage activity when activated by a target RNA in different model organisms [165] and is more effective and specific than RNAi in mammalian cells [166,167]. For this purpose, CRISPR-Cas13 has been proposed and used as a tool for SARS-CoV-2 detection [168]. Furthermore, Blanchard and co-workers found that using CRISPR/Cas13a mRNA specific for highly conserved regions of influenza virus and SARS-CoV-2, efficiently degraded influenza RNA in lung tissue when administered after infection, while in hamsters, Cas13a reduced replication of SARS-CoV-2 and reduced the symptoms [169]. Lastly, a promising example for CRISPR-Cas13 application is the study of Rashnonejad and co-workers against Facioscapulohumeral muscular dystrophy. They developed different Cas13b-gRNAs that target various Double Homeobox 4 (DUX4) mRNA parts and verified a decrease over 90% of DUX4 protein in treated cells. Moreover, cell viability was improved, as well as cell death prevention in vitro and in vivo [170]. Overall, the applications of the CRISPR-Cas13 in diagnostics are of interest and will open up new avenues for their in vivo applications, such as RNA knockdown and editing.

## 7. Conclusions and Future Directions

mRNA holds significant promise in gene therapy, in the control of emerging pandemic infectious diseases, and in diseases where no effective cure or treatment is available, avoiding several problems associated with therapies based on DNA or even based on proteins produced recombinantly. Applications such as immunotherapy and gene editing require protein expression only for limited periods of time, so transient protein expression by mRNA facilitates a wide range of biological processes without the risk of genomic integration. The main factors that prevent the clinical progression of mRNA therapy in chronic diseases are the transient expression and immunogenicity of the mRNA produced by IVT, in addition to the lack of sufficiently effective delivery systems to perform an effective transfection [1]. Although there are still limitations to the use of this biomolecule, over time these barriers have been overcome, through better knowledge of the biomolecule, modifications that can improve mRNA stability and technological evolution on the formulation of new, efficient, safe, and specific administration systems [29]. The development of a targeted non-viral delivery system for mRNA will be the culmination of years of research into therapies using this biomolecule. Finding the best targeting strategy is still the holy grail of nanomedicine. Indeed, vaccination using mRNA vaccines has enormous potential over conventional vaccines. This option offers benefits both in the speed of production and in the cost for developing the mRNA vaccine. Currently, mRNA-based vaccines promise to become the future and revolutionize the world of vaccination for therapeutic and prophylactic applications. The approval of the new mRNA vaccines for COVID-19 provides very good perspectives on the technology. It was an amazing achievement for the mRNA community. That said, the therapeutic use of mRNA has a huge potential to revolutionize medicine as we know it.

## Figures and Tables

**Figure 1 pharmaceutics-13-02090-f001:**
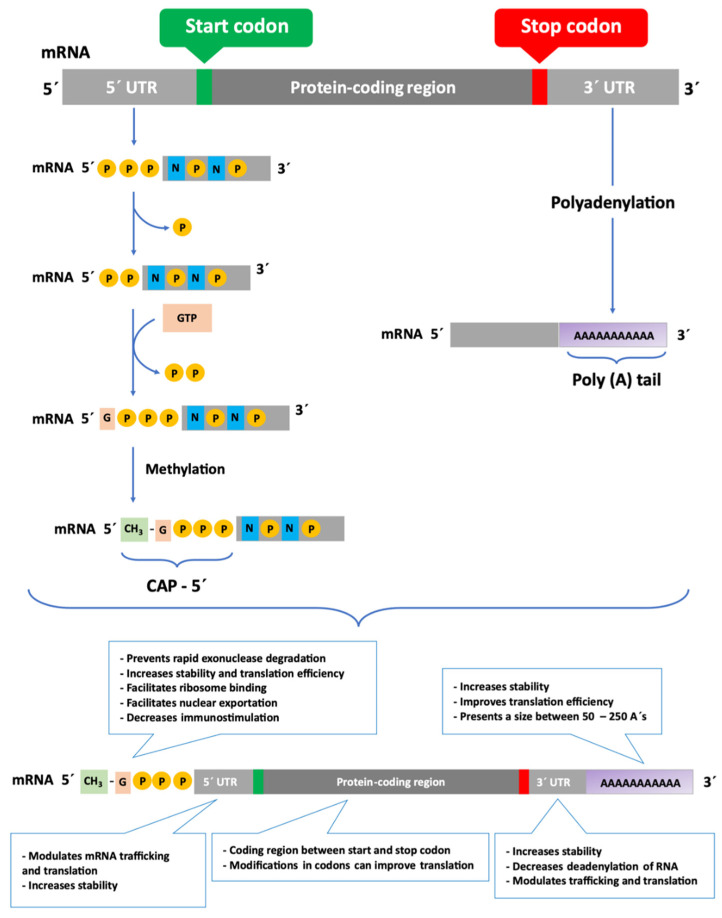
Schematic representation of eukaryotic mRNA, integrating the fundamental regions for its processing, such as 5′ UTR, protein coding region, and 3′ UTR. Schematization of polyadenylation process and introduction of Cap at the 5′ end, as well as its biological and structural functions. P—Phosphate group; N—Nucleotide; CH_3_—Methyl group; A—Adenine; G—Guanine; UTR—Untranslated region.

**Figure 2 pharmaceutics-13-02090-f002:**
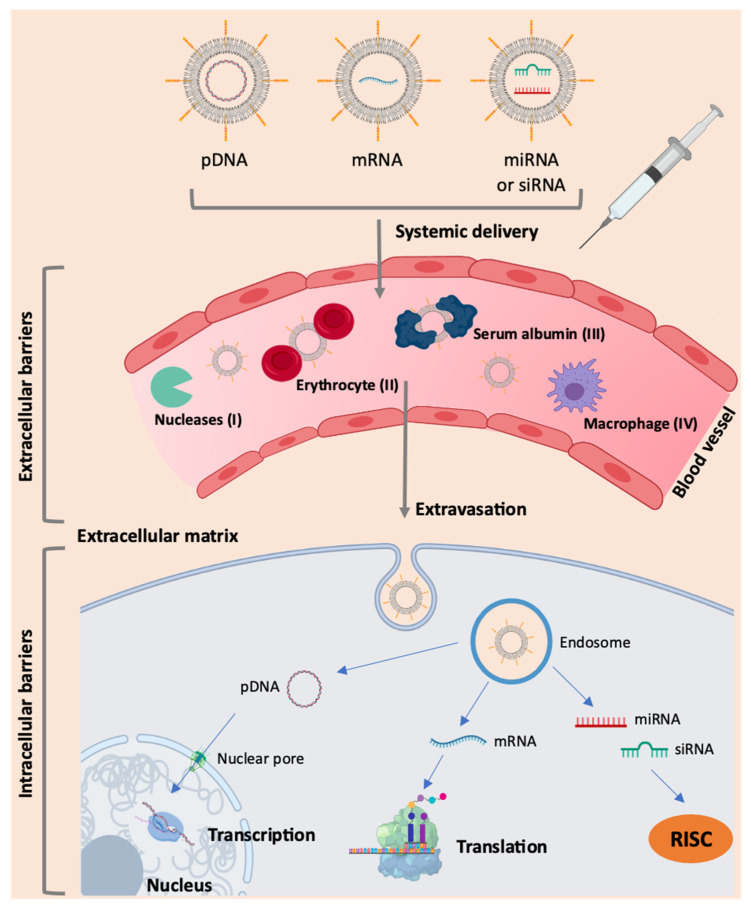
Schematic representation of Extra- and Intracellular Barriers for nucleic acids delivery. (I)—Endo and exonuclease degradation; (II)—Interaction and binding with erythrocytes; (III)—Binding and aggregation via serum protein complexation; (IV)—Immune activation to delivered nucleic acids.

**Figure 3 pharmaceutics-13-02090-f003:**
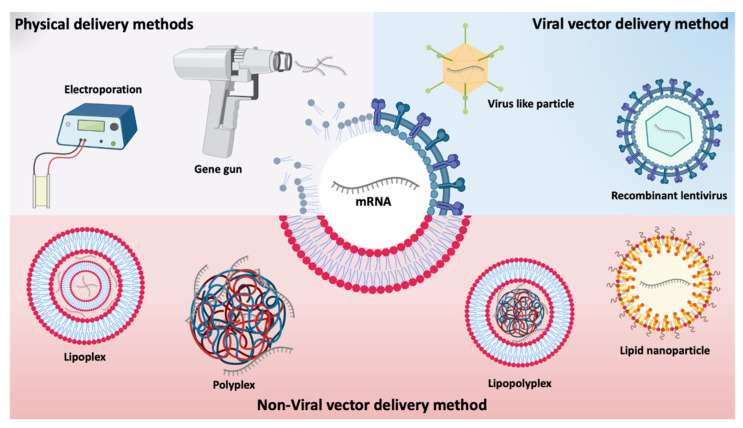
Schematic representation of different delivery strategies for mRNA therapies. There are represented 3 methods for the delivery of mRNA: physical methods, such as electroporation and gene gun; viral delivery systems, that uses recombinant virus; and non-viral delivery systems, such as lipid nanoparticles, polyplex, lipoplex and lipopolyplex.

**Figure 4 pharmaceutics-13-02090-f004:**
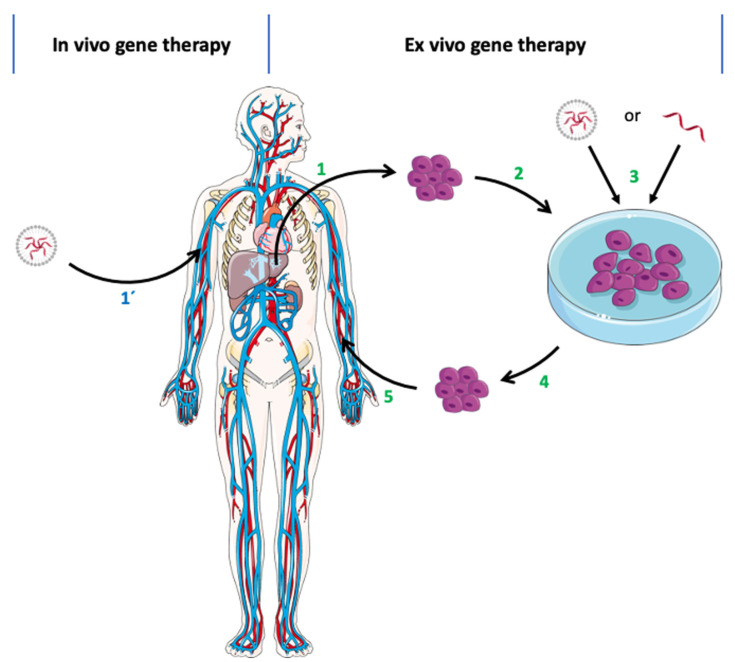
1 (Green)—Extraction of the patient’s cells by biopsy; 2—In vitro cell growth of cells from biopsy; 3—In vitro cell transfection by encapsulated and/or naked synthetic mRNA; 4—Cell growth of transfected cells and undergoing mRNA translation; 5—Autograft with mRNA transfected cells for therapeutic protein production. 1′ (Blue)—Systemic (intravenous or intramuscular) or localized (target organ or tissue) delivery of nanoparticle and mRNA formulations into the patient.

**Figure 5 pharmaceutics-13-02090-f005:**
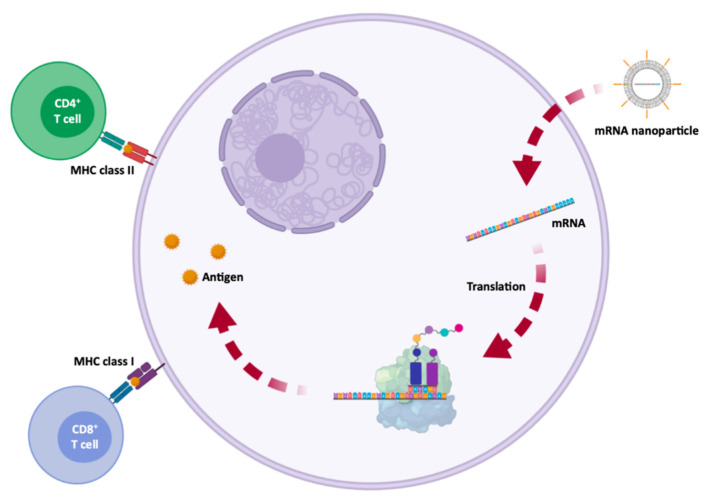
Antigen processing and presentation by dendritic cells, for adaptive immune system activation, following subcutaneous injection of a mRNA vaccine. A synthetic mRNA is internalized by antigen presenting dendritic cells, where the mRNA is translated. Then, the antigen is exposed by class I or II major histocompatibility complex (MHC) molecules and is later recognized by CD8^+^ or CD4^+^ T cells, activating chemical and humoral responses.

**Figure 6 pharmaceutics-13-02090-f006:**
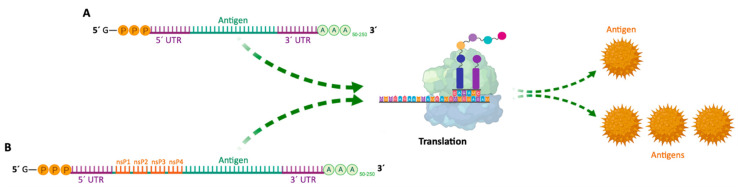
(**A**)—Schematic structure of conventional non-amplifying mRNA vaccine. (**B**)—Schematic structure of self-amplifying mRNA vaccine (replicon). UTR—Untranslated region; nsP—Non-structural proteins; A—Adenine; G—Guanine; P—Phosphate group.

**Figure 7 pharmaceutics-13-02090-f007:**
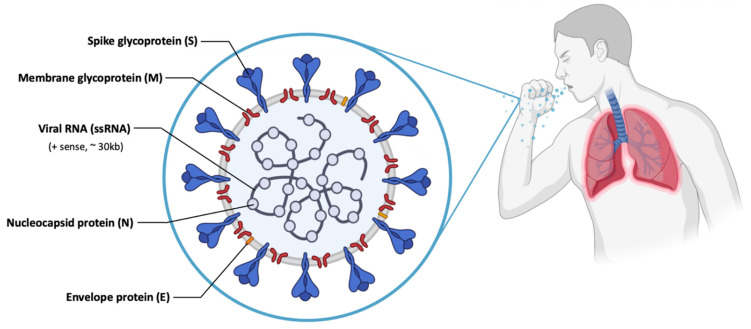
Scheme of the structure of SARS-CoV-2, with different viral proteins indicated.

**Figure 8 pharmaceutics-13-02090-f008:**
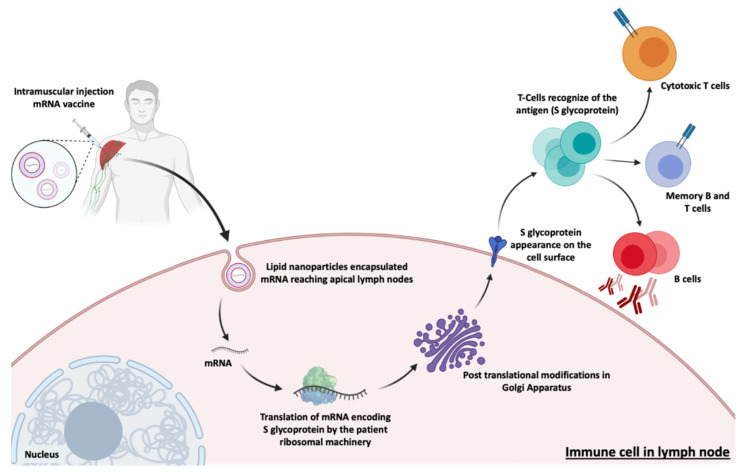
Actuation mechanism of the main mRNA vaccines against SARS-CoV-2. This process begins with the injection, in the patient’s deltoid muscle, of the mRNA usually encapsulated in lipid nanoparticles (LNPs). LNPs loaded with the mRNA encoding the SARS-CoV-2 spike glycoprotein (S), can reach the apical lymph nodes where they transfect dendritic cells. After entry and release of the cell endosome, the mRNA sequence is expressed and post-translational modifications occur. Subsequently, the S glycoprotein is transported and presented in the cell membrane of immune cells (antigen presenting cells). The S glycoproteins incites a specific cytotoxic and humoral immune response, leading to the production of antibodies against SARS-CoV-2, with the aim of achieving immunization against COVID-19.

**Figure 9 pharmaceutics-13-02090-f009:**
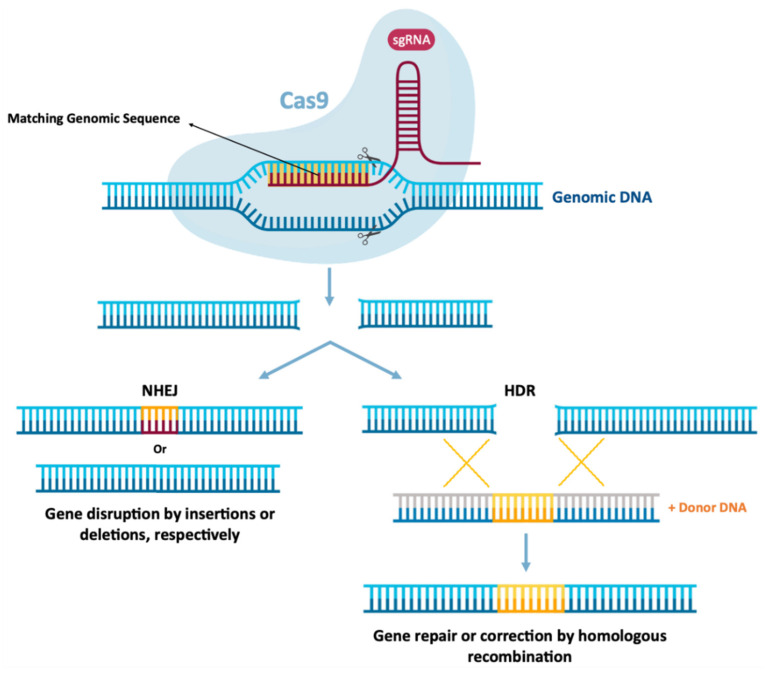
Schematic of CRISPR-Cas9-mediated genome editing. A CRISPR-Cas9 endonuclease is directed to a DNA sequence by means of a single guide RNA sequence (sgRNA), resulting in double strand cleavage. Subsequently they are repaired by non-homologous final union (NHEJ) or homology-directed repair (HDR). NHEJ repair provides errors, often leads to insertion or deletion mutations, which can lead to genome instability. Alternatively, in the presence of an exogenous donor DNA model, it can be repaired through error-free HDR, projecting precise DNA changes.

**Table 1 pharmaceutics-13-02090-t001:** Structural comparison of RNA and DNA molecules.

Characteristics	DNA 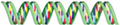	RNA 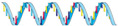
Type of sugar	Deoxyribose	Ribose
Bases	A, T, C, G	A, U, C, G
Double or single stranded	Double	Single
Secondary structure	Double helix	Many types
Stability	Stable	Easily degraded

**Table 2 pharmaceutics-13-02090-t002:** Therapeutic approaches based on mRNA and their functions.

Therapeutic Approach	Objective/Function
Protein Replacement	Restore function, increase expression or replace protein in rare monogenic diseases
Cell reprogramming	Modulate cellular behavior by expressing transcription and/or growth factors
Immunotherapies	Elicit specific immune responses against target cells, for example through therapeutic antibodies

**Table 3 pharmaceutics-13-02090-t003:** Advantages and disadvantages of non-amplifying mRNA Vaccines and SAM Vaccines.

mRNA Vaccines	Structure	Advantages	Disadvantages	References
Non-amplifying mRNA Vaccines	Basic structure of the mRNA, with a coding region for the desired antigens.	- Relatively small mRNA size (~2–3 kb). - Absence of additional proteins, minimizing unwanted immune interactions. - Relatively easy to produce and amplify. - Simplified sequence engineering. - Direct antigen expression.	- Potential toxicity from modified nucleotides. - Short duration of expression. - Need for high RNA doses. - Low antigen quantity.	[8,86]
SAM Vaccines	Encode a manipulated RNA virus genome (replicon). It generally contains two different protein coding regions, one encoding nonstructural proteins involved in mRNA capping and replication, and the other in antigen expression.	- High yield of target antigen. - Enhanced and prolonged antigen expression. - Lower effective RNA doses (more safe). - Intrinsic adjuvant effect. - Potential apoptosis of vaccine-carrying cells due to vaccine self-amplification (enhanced cross-presentation). - Option for single-vector delivery of multiple or complex antigens.	- RNA replicons are not able to tolerate many of the synthetic nucleotide modifications and sequence alterations. - Inclusion of unrelated proteins, which may increase unwanted immunogenicity. - Large replicon size (~10 kb), decreasing cell internalization efficiency. - Interaction between nsPs and host factors yet to be addressed. - Longer RNA length (more difficult production). - Potential elevated inflammation.	[8,13,29,86,93]

**Table 4 pharmaceutics-13-02090-t004:** Clinical trials for RNA-Based Protein Therapy (Protein replacement, cell reprogramming, immunotherapy) and gene editing.

Name	Therapetic Modality	Protein Target	Administration Method	Delivery Vehicle	Disease	Sponsor Institution	ClinicalTrials.gov Identifer	Phase	Therapeutic Approach	References
MRT5005	mRNA	CFTR	Inhalation	LNPs	Cystic fibrosis	Translate Bio	NCT03375047	I/II	Protein Replacement	[94]
AZD8601	mRNA	VEGF-A	Intracardiac injection	Naked mRNA	Heart failure	AstraZeneca	NCT03370887	II	Cell reprogramming	[15]
CV7201	mRNA	Rabies virus glycoprotein	I.D or I.M	RNActive, protamine	Rabies	CureVac	NCT02241135	I	Immunotherapy	[95]
CV7202	mRNA	Rabies virus glycoprotein	I.M	LNPs	Rabies		NCT03713086	I	Immunotherapy	[13]
CV9201	mRNA	TAAs	I.D	RNActive, protamine	NSCLC		NCT00923312	I/II	Immunotherapy	[96]
CV9202	mRNA	TAAs	I.D	RNActive, protamine	NSCLC		NCT03164772	I/II	Immunotherapy	[8]
CV9104	mRNA	TAAs	I.D	RNActive, protamine	Prostate carcinoma		NCT02140138	II	Immunotherapy	
HARE-40	mRNA	HPV antigen CD40	I.D	Naked RNA	HPV-driven squamous cell carcinoma	BioNTech	NCT03418480	I/II	Immunotherapy	
Lipo-MERIT	mRNA	TAAs: NYESO-1, MAGE-A3, tyrosinase, and TPTE	I.V	Lipo-MERIT, DOTMA(DOTAP)/ DOPE lipoplex	Advanced melanoma		NCT02410733	I	Immunotherapy	
IVAC	mRNA	3 TAAs selected from a warehouse and p53 RNA; Neo-Ag based on NGS screening	I.V	Lipo-MERIT, DOTMA(DOTAP)/DOPE lipoplex	TNBC	BioNTech	NCT02316457	I	Immunotherapy	[8]
RBL001/RBL002	mRNA	TAAs	Ultrasound guided I.N	Naked mRNA	Melanoma		NCT01684241	I	Immunotherapy	
IVAC MUTANOME	mRNA	Neo-Ag	Ultrasound guided I.N	Naked mRNA	Melanoma		NCT02035956	I	Immunotherapy	
RO7198457	mRNA	Neo-Ag	I.V	Naked mRNA	Melanoma; NSCLC; Bladder cancer		NCT03289962	I	Immunotherapy	
mRNA-1325	mRNA	Zika virus antigen	I.D	LNPs	Zika virus	Moderna	NCT03014089	I	Immunotherapy	
mRNA-1653	mRNA	hMPV and hPIV type 3 vaccine	I.D	LNPs	hMPV and hPIV infection		NCT03392389	I	Immunotherapy	
VAL-506440	mRNA	H10N8 antigen	I.D	LNPs	Influenza	Moderna	NCT03076385	I	Immunotherapy	[97]
VAL-339851	mRNA	H7 influenza antigen	I.D	LNPs	Influenza		NCT03345043	I	Immunotherapy	
mRNA-1647/1443	mRNA	CMV glycoprotein H pentamer complex	I.D	LNPs	CMV infection		NCT03382405	I	Immunotherapy	[8]
mRNA-2416	mRNA	Human OX40L	I.D	LNPs	Solid tumor malignancies or lymphoma		NCT03323398	I	Immunotherapy	
mRNA-4157	mRNA	Neo-Ag	Intratumoral	LNPs	Solid tumor		NCT03313778	I	Immunotherapy	
mRNA-4650	mRNA	Neo-Ag	I.M	Naked mRNA	Melanoma; Colon cancer; GI cancer; Genitourinary cancer; HCC		NCT03480152	I/II	Immunotherapy	[98,99]
mRNA-1388	mRNA	VAL-181388	I.M	LNPs	CHIKV		NCT03325075	I	Immunotherapy	[12]
mRNA-2752	mRNA	OX40L, IL-23, and IL-36γ	Intratumoral	LNPs	Solid tumor or lymphoma	Moderna/AstraZeneca	NCT03739931	I	Immunotherapy	[13]
iHIVARNA-01	mRNA	Trimix (CD40L, CD70 and caTLR4 RNA—mRNA-transfected)	I.N	Naked mRNA	HIV infection	Hospital Clínic de Barcelona	NCT02413645	I	Immunotherapy	[100]
	mRNA		I.N	Naked mRNA	HIV infection	Erasmus Medical Center	NCT02888756	II	Immunotherapy	[101]
-	mRNA	CT7, MAGE-A3, and WT1 mRNA-electroporated LCs	I.D	DC-loaded mRNA	Malignant melanoma	Memorial Sloan Kettering Cancer Center	NCT01995708	I	Immunotherapy	[8]
-	mRNA	HIV-1 Gag- and Nef-transfected DCs	I.D	DC-loaded mRNA	HIV infection	Massachusetts General Hospital	NCT00833781	I/II	Immunotherapy	[102]
-	mRNA	Neo-Ag	S.C	Naked mRNA	Solid tumor malignancies or lymphoma	Changhai Hospital Stemirna Therapeutics	NCT03468244	N.A	Immunotherapy	[8]
-	mRNA	TAA for melanoma (Melan-A, MAGE-A1, MAGE-A3, survivin, GP100, and tyrosinase)	I.D	Naked mRNA	Melanoma	University Hospital Tuebingen	NCT00204516	I/II	Immunotherapy	[12]
-	mRNA	TAA-transfected DC	I.D or I.N	DC-loaded mRNA	Malignant melanoma	Oslo University Hospital	NCT01278940	I/II	Immunotherapy	[103]
-	mRNA		I.D	DC-loaded mRNA	Prostate cancer		NCT01278914	I/II	Immunotherapy	[8]
AVX601	Replicon	Alphavirus replicon vaccine expressing CMV genes	I.M or S.C	-	CMV	AlphaVax	NCT00439803	I	Immunotherapy	[8]
AVX502	Replicon	Alphavirus replicon vaccine expressing an influenza HA protein	I.M or S.C	-	Influenza		NCT00440362; NCT00706732	I/II	Immunotherapy	
AVX101	Replicon	Alphavirus replicon, HIV-1 subtype C Gag vaccine	I.M or S.C	-	HIV infections		NCT00097838; NCT00063778	I	Immunotherapy	[104]
AVX701	Replicon	Alphavirus replicon encoding the protein	I.M or S.C	-	Colon cancer; CRC; Breast cancer; Lung cancer; Pancreatic cancer		NCT01890213; NCT00529984	I/II	Immunotherapy	[8]
NY-ESO-1	CRISPR-Cas9	PD-1 and TCR	Ex vivo	Autologous T cells	Multiple myeloma; Synovial sarcoma; Melanoma	University of Pennsylvania	NCT03399448	I	Gene Editing	[8]
CRISPR/TALEN-HPV E6/E7	CRISPR/Cas9, TALEN	E6 and E7	N.A	Plasmid DNA in gel	Cervical intraepithelial neoplasia	First Affiliated Hospital, Sun Yat-Sen University	NCT03057912	I	Gene Editing	[105,106]
CTX001	CRISPR-Cas9	BCL11A	Ex vivo	Modified CD34^+^ hHSPCs	ß-thalassemia	Vertex Pharmaceuticals Incorporated	NCT03655678	I/II	Gene Editing	[8]
-	CRISPR-Cas9	PD-1 and TCR	Ex vivo	CAR-T cells	Mesothelin positive multiple solid tumors	Chinese PLA General Hospital	NCT03545815	I	Gene Editing	
-	CRISPR-Cas9	CD19 and CD20	Ex vivo	Dual specificity CAR-T cells	ß cell leukemia and lymphoma		NCT03398967	I/II	Gene Editing	
UCART019	CRISPR-Cas9	CD19	Ex vivo	CAR-T cells	ß cell leukemia and lymphoma		NCT03166878	I/II	Gene Editing	[107]
-	CRISPR-Cas9	PD-1	Ex vivo	Cytotoxic T lymphocytes	EBV-associated malignancies	Yang Yang	NCT03044743	I/II	Gene Editing	[108,109]
SB-728mR-HSPC	ZFN mRNA	CCR5	Ex vivo (mRNA)	CD34^+^ hHSPCs	HIV	City of Hope Medical Center	NCT02500849	I	Gene Editing	[110]
SB-728mR-T	ZFN mRNA	CCR5	Ex vivo (mRNA)	T cells	HIV	Sangamo Therapeutics	NCT02225665	I/II	Gene Editing	[111]

CFTR—Cystic fibrosis transmembrane conductance regulator; LNPs—Lipid nanoparticles; VEGF-A—Vascular endothelial growth factor A; I.D.—Intradermal; I.M.—Intramuscular; TAAs—Tumor-associated antigens; NSCLC—Non-small-cell lung carcinoma; HPV—Human Papillomavirus; I.N.—Intranodal; MAGE-A—Melanoma-associated antigen-A; TPTE—Putative tyrosine-protein phosphatase; I.V—Intravenous; Neo-Ag—Neo-antigen; NGS—Next-Generation Sequencing; TNBC—Triple-negative breast cancer; hMPV—Human metapneumovirus; hPIVs—Human parainfluenza viruses; CMV—Cytomegalovirus; GI—Gastrointestinal; HCC—Hepatocellular cancer; CHIKV—Chikungunya virus; IL—Interleukin; HIV—Human immunodeficiency virus; WT1—Wilms’ tumor 1; LCs—Langerhans cells; DC—Dendritic cell; S.C.—Subcutaneous; N.A—Not applicable; GP100—Glycoprotein 100; HA—Hemagglutinin; CRC—Colorectal cancer; PD-1—Programmed cell death protein 1; TLR—Toll-like receptor; BCL11A—B-cell lymphoma/leukemia 11A; hHSPCs—Human hematopoietic stem and progenitor cells; CAR—Chimeric antigen receptor; EBV—Epstein-Barr virus; ZFN—Zinc-finger nucleases; CCR5—C-C Motif Chemokine Receptor 5.

**Table 5 pharmaceutics-13-02090-t005:** LNP carriers of the COVID-19 mRNA vaccines (lipidic constituents) [130,131].

Lipid Name	Role	Abbreviation	Molar Lipid Ratios (%) (Ionizable Cationic Lipid:Neutral Lipid:Cholesterol:PEG-ylated Lipid)
BNT162b2 vaccine			
4-hydroxybutyl)azanediyl)bis(hexane-6,1-diyl)bis(2-hexyldecanoate	ionizable cationic lipid	ALC-0315	46.3:9.4:42.7:1.6
1,2-Distearoyl-sn-glycero-3-phosphocholine	helper lipid	DSPC	
cholesterol	helper lipid	Chol	
2-[(polyethylene glycol)-2000]-*N*,*N*-ditetradecylacetamide	PEG-lipid	ALC-0159	
mRNA-1273 vaccine			
heptadecan-9-yl 8-((2-hydroxyethyl)[6-oxo-6-(undecyloxy)hexyl]amino)octanoate	ionizable cationic lipid	SM-102	50:10:38.5:1.5
1,2-distearoyl-sn-glycero-3-phosphocholine	helper lipid	DSPC	
cholesterol	helper lipid	Chol	
1,2-Dimyristoyl-rac-glycero-3-methoxypolyethylene glycol-2000	PEG-lipid	PEG2000-DMG	

**Table 6 pharmaceutics-13-02090-t006:** mRNA vaccines and new candidates for COVID-19 [150].

Name	Therapetic Modality	Protein Target	Administration Method	Delivery Vehicle	Developer	ClinicalTrials.gov Identifer, EU Clinical Trials Register or Chinese Clinical Trial Register	Phase
mRNA-1273	mRNA	Spike Glycoprotein	Intramuscular	LNP	Moderna/NIAID	EUCTR2021-002327-38-NL	IV
BNT162b2	mRNA	RBD/Spike Glycoprotein	Intramuscular	LNP	Pfizer/BioNTech + Fosun Pharma	NCT04760132	IV
CVnCoV Vaccine	mRNA	Spike Glycoprotein	Intramuscular	LNP	CureVac AG	NCT04674189	III
ARCT-021	mRNA	Spike Glycoprotein	Intramuscular	LNP	Arcturus Therapeutics	NCT04668339	II
LNP-nCoVsaRNA	mRNA	Spike Glycoprotein	Intramuscular	LNP	Imperial College London	ISRCTN17072692	I
SARS-CoV-2 mRNA vaccine (ARCoV)	mRNA	RBD	Intramuscular	LNP	AMS/Walvax Biotechnology and Suzhou Abogen Biosciences	NCT04847102	III
ChulaCov19 mRNA vaccine	mRNA	Spike Glycoprotein	Intramuscular	LNP	Chulalongkorn University	NCT04566276	I
PTX-COVID19-B, mRNA vaccine	mRNA	Spike Glycoprotein	Intramuscular	LNP	Providence therapeutics	NCT04765436	I
saRNA formulated in a NLC	mRNA	-	-	NLC	Infectious Disease Research Institute/Amyris, Inc.	-	Pre-Clinical
LNP-encapsulated mRNA encoding S	mRNA	Spike Glycoprotein	-	LNP	Max-Planck-Institute of Colloids and Interfaces	-	Pre-Clinical
Self-amplifying RNA	mRNA	-	-	-	Gennova	-	Pre-Clinical
mRNA	mRNA	-	-	-	Selcuk University	-	Pre-Clinical
LNP-mRNA	mRNA	-	-	LNP	Translate Bio/Sanofi Pasteur	-	Pre-Clinical
LNP-mRNA	mRNA	-	-	LNP	CanSino Biologics/Precision NanoSystems	-	Pre-Clinical
LNP-encapsulated mRNA cocktail encoding VLP	mRNA	-	-	LNP	Fudan University/Shanghai JiaoTong University/RNACure Biopharma	-	Pre-Clinical
LNP-encapsulated mRNA encoding RBD	mRNA	RBD	-	LNP	Fudan University/Shanghai JiaoTong University/RNACure Biopharma	-	Pre-Clinical
Replicating Defective SARS-CoV-2 derived RNAs	mRNA	-	-	-	Centro Nacional Biotecnología (CNB-CSIC), Spain	-	Pre-Clinical
LNP-encapsulated mRNA	mRNA	-	-	LNP	University of Tokyo/Daiichi-Sankyo	-	Pre-Clinical
Liposome-encapsulated mRNA	mRNA	-	-	LNP	BIOCAD	-	Pre-Clinical
Several mRNA candidates	mRNA	-	-	-	RNAimmune, Inc.	-	Pre-Clinical
mRNA	mRNA	-	-	-	FBRI SRC VB VECTOR, Rospotrebnadzor, Koltsovo	-	Pre-Clinical
mRNA	mRNA	-	-	-	China CDC/Tongji University/Stermina	-	Pre-Clinical
mRNA in an intranasal delivery system	mRNA	-	Intranasal	-	eTheRNA	-	Pre-Clinical
mRNA	mRNA	-	-	-	Greenlight Biosciences	-	Pre-Clinical
mRNA	mRNA	-	-	-	IDIBAPS-Hospital Clinic, Spain	-	Pre-Clinical
mRNA	mRNA	-	-	-	Providence Therapeutics	-	Pre-Clinical
mRNA	mRNA	-	-	-	Cell Tech Pharmed	-	Pre-Clinical
mRNA	mRNA	-	-	-	ReNAP Co.	-	Pre-Clinical
D614G variant LNP-encapsulated mRNA	mRNA	-	-	LNP	Globe Biotech Ltd.	-	Pre-Clinical
Encapsulated mRNA	mRNA	-	-	-	CEA	-	Pre-Clinical

LNPs—Lipid nanoparticles; NIAID—National Institute of Allergy and Infectious Diseases; RBD—Receptor-binding domain; AMS—Academy of Military Science.

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
