# Peer review of "mRNA, a Revolution in Biomedicine"

_pharmaceutics, 2021, doi:10.3390/pharmaceutics13122090_

Round 1

Reviewer 1 Report

Review of Baptista et al.,

Titled: mRNA, a Revolution in Biomedicine

For: Pharmaceutics

The review manuscript of Baptista et al. described broadly the mRNA field. The authors cover in length (too much length in my opinion)the history of mRNA, mRNA versus DNA, vehicles use for mRNA delivery, Therapeutic use of mRNA, the use of mRNA in protein replacement, the use of mRNA in fighting covid 19 pendemic, and more.

Sometimes less is more.  I think this review should cover only the mRNA biology and not the therapeutic part, as the therapeutic part is poorly presented.

The authors totally missed the main point in mRNA therapeutic, which is that mRNA therapeutic dose not work!!!

You need to modify it to make it work. None of clinical trails using self-replicate or mRNA were successful (ask Curevac), only the one that modified the uridine to pseudouridine worked (modeRNA and BioNTech)!!

uridine is mention only one time in the text !!!

pseudouridine is not mention at all!! This is similar of written about trip in Paris without mentioning the Eiffel tower!!

As an expert in this field is my humble opinion that this review manuscript needs to be truncated dramatically and discus only mRNA biology (without therapeutic) or if therapeutic should be mention, it should be focused on the differences between modified mRNA or mRNA therapeutic.

Author Response

Reviewer #1
The review manuscript of Baptista et al. described broadly the mRNA field. The authors cover in length (too much length in my opinion) the history of mRNA, mRNA versus DNA, vehicles use for mRNA delivery, Therapeutic use of mRNA, the use of mRNA in protein replacement, the use of mRNA in fighting covid 19 pendemic, and more. Sometimes less is more. I think this review should cover only the mRNA biology and not the therapeutic part, as the therapeutic part is poorly presented. The authors totally missed the main point in mRNA therapeutic, which is that mRNA therapeutic dose not work!!! You need to modify it to make it work. None of clinical trails using self-replicate or mRNA were successful (ask Curevac), only the one that modified the uridine to pseudouridine worked (modeRNA and BioNTech)!! uridine is mention only one time in the text !!! pseudouridine is not mention at all!! This is similar of written about trip in Paris without mentioning the Eiffel tower!! As an expert in this field is my humble opinion that this review manuscript needs to be truncated dramatically and discus only mRNA biology (without therapeutic) or if therapeutic should be mention, it should be focused on the differences between modified mRNA or mRNA therapeutic.
Answer:
We thank the reviewer for analyzing our manuscript. Our goal is to give an overview of both mRNA biology and its use in biomedicine. We agree that it is difficult to cover all aspects and we are sorry for have missed the discussion on mRNA modifications etc. We agree with  the Reviewer’s comment and have included a discussion concerning the need for mRNA modifications, when considering the therapeutic application. The discussion also includes a comparison of Pfizer/BioNTech and Moderna with CureVac vaccines (please see page 26-27, lines 646-715).

Reviewer 2 Report

The authors present a review on mRNA, covering fundamental ideas through to current development towards clinical therapeutic use. Many reviews have been written on mRNA (and indeed, many are cited in this text), and the novel aspect to this review is incorporating the recent development of mRNA vaccines and therapeutics, particularly with regards to COVID-19 treatment. However many of the topics covered in this review are simplistic and weighted heavily towards explaining the tenets of a basic mechanism rather than highlighting primary and state-of-the-art research. As such, further depth needs to be added to the text before the review is suitable for publication, in addition to the technical aspects of the writing and language used, which require improvement.

First, there is an overreliance here on citing from reviews rather than primary sources when describing fundamental mechanisms. Very little on the first two pages is from primary sources: most statements on pages one and two cite reference [2], a book on genetics, and the majority of references up to [20] are reviews. The authors should find the primary literature where each mechanism, material, sequence etc discovery was made and cite those original works instead.

The authors discuss concepts such as polyadenylation only briefly across different sections of the review, when there has been great improvement in this technology recently: for example, a segmented poly(A) tail was utilised in the BNT162b2 mRNA vaccine towards improving its stability and translation capacity in target cells. There is also a lack of discussion on mRNA development strategies such as codon optimisation, synthetic bases (e.g. 5moU and N1-methylpseudouridine) and use of specific sequences to improve mRNA performance such as the Kozak sequence, all of which are very relevant to mRNAs in both current therapeutic use and in clinical development. These topics should all be addressed and expanded upon. 

Specific lines/areas to address in the manuscript:

  • P2, L41, "an astonishing relevance" is too informal. More cases of inappropriately informal language (e.g. use of "for sure" in the Conclusions section) and subjective opinions of the authors not supported by cited evidence (e.g. P1, L27, regarding DNA, "Scientists changed their target" -- research is still being undertaken on DNA?) should be revised.
  • P1, L30: "(hairpins, loops,...)" is there an unfinished thought here?
  • P3, L89: Figure 1 obscures the final sentence of this paragraph. 
  • Challenges in mRNA delivery section: challenges/disadvantages in the use of non-viral transporters to deliver mRNA are not discussed. For example, though PEI may be "the most frequently used cationic polymer for nucleotide delivery" it has seen very little success when utilised in vivo.
  • Cap-5' and CAP-5' are used interchangeably throughout the manuscript.
  • Table 5: 'glicoprotein' should be spelled 'glycoprotein'. Furthermore, this table has a number of missing sections that need to be filled in; for example, it is known that CVnCoV and ARCoV are delivered in LNPs, but this is not specified in the vehicle row.
  • Section 6.1, 'Protein replacement therapies', is too long and details mechanisms that are not protein replacement, for example immunotherapies. These should be split into different sections. Additionally, the structure and pathology of Sars-CoV-2 is beyond the scope of this review and should only be briefly summarised. This is a review on mRNA: in this section more depth should be added on the mRNA vaccines against COVID-19 themselves, i.e. Moderna and Pfizer's candidates. In particular, discussion of CureVac's failed candidate would be interesting and add more novelty to your review, given that the design of the mRNA has been postulated as its key point of failure.
  • Section 6.2, 'Gene editing': there have been several papers demonstrating successful use of mRNA-CRISPR systems (Feng Zhang's recent work comes to mind, i.e. doi:10.1073/pnas.2020401118) and even a commercial product from GeneArt, but none are actually discussed here.
  • References: style is inconsistent and should be ameliorated.

Author Response

Reviewer #2

The authors present a review on mRNA, covering fundamental ideas through to current development towards clinical therapeutic use. Many reviews have been written on mRNA (and indeed, many are cited in this text), and the novel aspect to this review is incorporating the recent development of mRNA vaccines and therapeutics, particularly with regards to COVID-19 treatment. However, many of the topics covered in this review are simplistic and weighted heavily towards explaining the tenets of a basic mechanism rather than highlighting primary and state-of-the-art research. As such, further depth needs to be added to the text before the review is suitable for publication, in addition to the technical aspects of the writing and language used, which require improvement.

Comments and Answers:

  1. Very little on the first two pages is from primary sources: most statements on pages one and two cite reference [2], a book on genetics, and the majority of references up to [20] are reviews. The authors should find the primary literature where each mechanism, material, sequence etc discovery was made and cite those original works instead.

Considering the suggestion of the Reviewer, which is very acknowledged, the authors provided additional references to original works. For example, regarding delivery systems, in topic 5, new original works are referenced (page 6-10, line 233-391). Other examples can be found in page 12, line 442-449 and in page 14, line 525-531. Regarding RNA modifications, in topic 6.2 (page 27, line 682-715) other original works were referenced. Also, novel examples of original papers were referred in page 35 line 58-60 and lines 68-94, and page 35-36, lines 100-112.

  1. There is also a lack of discussion on mRNA development strategies such as codon optimisation, synthetic bases (e.g. 5moU and N1-methylpseudouridine) and use of specific sequences to improve mRNA performance such as the Kozak sequence, all of which are very relevant to mRNAs in both current therapeutic use and in clinical development. These topics should all be addressed and expanded upon.

The authors totally agree with the Reviewer, and as suggested, additional information regarding the mRNA modification approaches was added to the manuscript (please see page 27, line 682-715).

  1. P2, L41, "an astonishing relevance" is too informal. More cases of inappropriately informal language (e.g. use of "for sure" in the Conclusions section) and subjective opinions of the authors not supported by cited evidence (e.g. P1, L27, regarding DNA, "Scientists changed their target" -- research is still being undertaken on DNA?) should be revised.

According to the Reviewer's comment, the language was revised to eliminate the subjective opinions.

  1. P1, L30: "(hairpins, loops,...)" is there an unfinished thought here?

The authors apologize for this misspelling error, which was corrected (please see page 1, line 30).

  1. P3, L89: Figure 1 obscures the final sentence of this paragraph.

The authors apologize for this formatting error, which was corrected (please see page 3, line 94).

  1. Challenges in mRNA delivery section: challenges/disadvantages in the use of non-viral transporters to deliver mRNA are not discussed. For example, though PEI may be "the most frequently used cationic polymer for nucleotide delivery" it has seen very little success when utilised in vivo.

The authors thank the Reviewer for this important comment. To better discuss this topic, the corresponding section was deeply restructured (please see pages 6-10, lines 238-399).

  1. Cap-5' and CAP-5' are used interchangeably throughout the manuscript.

The authors apologize for this inconsistency. The expression was uniformed to “Cap-5’”, throughout the manuscript.

  1. Table 5: 'glicoprotein' should be spelled 'glycoprotein'. Furthermore, this table has a number of missing sections that need to be filled in; for example, it is known that CVnCoV and ARCoV are delivered in LNPs, but this is not specified in the vehicle row.

The authors would like to apologize for the misspelling error. Furthermore, the authors thank the Reviewer for the suggestion and have included more specific information in Table 5 (now Table 6).

  1. Section 6.1, 'Protein replacement therapies', is too long and details mechanisms that are not protein replacement, for example immunotherapies. These should be split into different sections. Additionally, the structure and pathology of Sars-CoV-2 is beyond the scope of this review and should only be briefly summarised. This is a review on mRNA: in this section more depth should be added on the mRNA vaccines against COVID-19 themselves, i.e. Moderna and Pfizer's candidates. In particular, discussion of CureVac's failed candidate would be interesting and add more novelty to your review, given that the design of the mRNA has been postulated as its key point of failure.

The authors thank the Reviewer for this important comment, and have actually separated the information into two topics (please see page 12). Besides, we included additional information in the Protein Replacement topic (see page 12, lines 429-435 and lines 442-449). The structure of SARS-CoV-2 is summarized in Figure 7, and briefly explained on page 24, in accordance with the Reviewer’s suggestion. Information regarding SARS-CoV-2 was also included which is summarized on page 24 and 25. Furthermore, more information about Pfizer/BioNTech and Moderna vaccines was included, as well as a discussion on CureVac unsuccess (please see pages 25-27).

  1. Section 6.2, 'Gene editing': there have been several papers demonstrating successful use of mRNA-CRISPR systems (Feng Zhang's recent work comes to mind, i.e. doi:10.1073/pnas.2020401118) and even a commercial product from GeneArt, but none are actually discussed here.

The authors appreciate and agree with the suggestion made. Given this, the examples suggested were included, in order to improve the manuscript (see page 35-36, lines 68-112).

 References: style is inconsistent and should be ameliorated.

The authors appreciate the warning and references were reformulated and uniformized.

Reviewer 3 Report

The manuscript by Baptist et al. is an informative work on the last advances of the use of mRNA as a therapeutic tool. However, the manuscript should be improved for publication. My main comments are the following:

  1. In lane 30 something was left unfinished after loops.....
  2. In section structure and biogenesis, mRNA transcription (by RNA pol II) in eukaryotes should be described. 
  3. mRNA delivery strategies should be described deeply, especially those used by Moderna and Pfizer. As long as possible the composition of the nanoparticles used for mRNA delivery should be described in detail, since this is quite important.
  4. It is not clear from the text whether or not Moderna and Pfizer have used the full lenght S protein or only fragments of it as immunogen.

Author Response

Dear Reviewer,
First of all, the authors would like to acknowledge the reviewers for its careful analysis and constructive scientific comments provided on the manuscript entitled “mRNA, a Revolution in Biomedicine”. Those remarks and recommendations improved a lot our review. The authors have revised the manuscript and all the recommended modifications were made. All changes made to the text are marked.
We hope that the revised manuscript is meeting the requirements 
Best regards
Chantal PICHON

Reviewer #3 The manuscript by Baptist et al. is an informative work on the last advances of the use of mRNA as a therapeutic tool. However, the manuscript should be improved for publication.
Comments: 1. In lane 30 something was left unfinished after loops... The authors apologize for this misspelling error, which was corrected (please see page 1, line 30).
2. In section structure and biogenesis, mRNA transcription (by RNA pol II) in eukaryotes should be described. The authors thank the Reviewer’s suggestion, and included this important information in page 2, lines 56-58.
3. mRNA delivery strategies should be described deeply, especially those used by Moderna and Pfizer. As long as possible the composition of the nanoparticles used for mRNA delivery should be described in detail, since this is quite important. The authors thank the suggestion, and given the relevance of this theme, the authors totally reformulated topic 5. We made efforts to improve the description of mRNA delivery strategies, including various original works in this topic (page 6-10, line 238-399-394, page 25, line 632-645 and page 26, Table 5).
4. It is not clear from the text whether or not Moderna and Pfizer have used the full lenght S protein or only fragments of it as immunogen. The authors recognize this missing information, which was now included (please see page 27, lines 627-670).

Round 2

Reviewer 1 Report

I have no further questions to the authors

Author Response

Reviewer #1

I have no further questions to the authors.

The authors acknowledge the contribution made by the Reviewer and also thank the careful reading.

Reviewer 2 Report

The authors have addressed most major points and greatly improved the quality of the manuscript, however a few things still need to be amended.

One comment that has not been addressed is regarding a long section detailing the history, structure and pathology of Sars-CoV-2. In their response to my suggestion that this is beyond the scope of their review and should only be briefly summarised, the authors stated: "The structure of SARS-CoV-2 is summarized in Figure 7, and briefly explained on page 24, in accordance with the Reviewer’s suggestion. Information regarding SARS-CoV-2 was also included which is summarized on page 24 and 25." To claim my suggestion has been addressed is false -- the authors have not cut down any of the text in this section, and have in fact made it longer. I strongly advise that the majority of this text is cut. A couple of sentences would be sufficient to introduce COVID-19 and summarise its impact; the full history of the pandemic, specific symptoms and pathology of COVID-19 and the detailed structure of Sars-CoV-2 as written herein are not relevant to mRNA. The paragraph on Spike protein, for example, is okay to include because it relates back to mRNA, i.e. that the majority of mRNA vaccines target this protein. Furthermore, I would suggest integrating the paragraphs at P33, L1-12 into this section to lead into the section on the Moderna and Pfizer vaccines, as it would better suit the flow of the text.

One note on the newly added section regarding Moderna and Pfizer vaccines -- some information here might be outdated, or region-specific (i.e. if the dose prices and approvals are for the US, this should be specified). For example, while the authors claim Moderna's vaccine is only approved for those over 18 and Pfizer's over 16, Comirnaty (Pfizer) has been approved for ages 12 and over in the US since May and Spikevax (Moderna) has actually been approved for ages 12 and up in Australia (also worth specifying in the manuscript that the vaccines are now branded under those names).

Though the authors have addressed my comment regarding adding mRNA-CRISPR systems to the 'Gene editing' section, they might also consider mentioning CRISPR-Cas13, as this system specifically edits RNA (including mRNA) and its inclusion will further enrich the review.

Lastly, a number of structural and grammatical errors in certain sections of the text require fixing (examples given here are not exhaustive -- the authors should do a thorough screening of the paper to find and correct additional errors):  

P5, L179-180: "Since mRNA has been exposed in a very positive way to therapeutic actions in numerous diseases, the challenges for mRNA production started to be addressed in a more intensive way, in order to solve this problematic." This sentence is long and makes little sense (did the authors mean to write 'problem' here?), and would probably be easier just to cut out entirely. 

P8, L280: "Physical methods are simple and consist in employing ..." -- 'in' is incorrect here, 'of' should be used. ('Consists in' is repeated in P14, L512 and P26, L651 and 656; these should also be fixed).

P9, L306-7: "Different types of non-viral systems (polymeric, lipid and hybrid systems) have evolved to protect and improve the delivery of mRNA being mainly exploited the [43]." -- unfinished sentence, also 'evolved' is not correct in this context, as these are synthetic materials and not biological organisms (consider using 'developed' instead).

P10, L357-361 is a run-on sentence and should be revised. 

P27, L315: 'Kozak' is misspelled here.

Author Response

Reviewer #2

The authors have addressed most major points and greatly improved the quality of the manuscript, however a few things still need to be amended.

Comments and Answers:

  1. One comment that has not been addressed is regarding a long section detailing the history, structure and pathology of Sars-CoV-2. In their response to my suggestion that this is beyond the scope of their review and should only be briefly summarised, the authors stated: "The structure of SARS-CoV-2 is summarized in Figure 7, and briefly explained on page 24, in accordance with the Reviewer’s suggestion. Information regarding SARS-CoV-2 was also included which is summarized on page 24 and 25." To claim my suggestion has been addressed is false -- the authors have not cut down any of the text in this section, and have in fact made it longer. I strongly advise that the majority of this text is cut. A couple of sentences would be sufficient to introduce COVID-19 and summarise its impact; the full history of the pandemic, specific symptoms and pathology of COVID-19 and the detailed structure of Sars-CoV-2 as written herein are not relevant to mRNA. The paragraph on Spike protein, for example, is okay to include because it relates back to mRNA, i.e. that the majority of mRNA vaccines target this protein. Furthermore, I would suggest integrating the paragraphs at P33, L1-12 into this section to lead into the section on the Moderna and Pfizer vaccines, as it would better suit the flow of the text.

The authors deeply apologize for the misinterpretation of the previous suggestion from the Reviewer. The authors removed irrelevant information regarding SARS-CoV-2 history, specific symptoms and COVID-19 pathology (see page 24, lines 573-598).

Concerning the Reviewer suggestion, the authors moved the paragraphs at page 33, lines 1-12, to the section on the Moderna and Pfizer vaccines, which certainly improved the flow of the text (see pages 24-25, lines 599-626).

  1. One note on the newly added section regarding Moderna and Pfizer vaccines -- some information here might be outdated, or region-specific (i.e. if the dose prices and approvals are for the US, this should be specified). For example, while the authors claim Moderna's vaccine is only approved for those over 18 and Pfizer's over 16, Comirnaty (Pfizer) has been approved for ages 12 and over in the US since May and Spikevax (Moderna) has actually been approved for ages 12 and up in Australia (also worth specifying in the manuscript that the vaccines are now branded under those names).

The authors totally agree with the Reviewer point of view. Therefore, information regarding vaccines approval was updated and correctly cited according to US approvals, by FDA (please see page 26, lines 650-660).

  1. Though the authors have addressed my comment regarding adding mRNA-CRISPR systems to the 'Gene editing' section, they might also consider mentioning CRISPR-Cas13, as this system specifically edits RNA (including mRNA) and its inclusion will further enrich the review.

According to the Reviewer's comment, the authors added information concerning CRISPR-Cas13 (please see page 35, lines 92-113), which greatly enriched the manuscript.

  1. Lastly, a number of structural and grammatical errors in certain sections of the text require fixing (examples given here are not exhaustive -- the authors should do a thorough screening of the paper to find and correct additional errors):

P5, L179-180: "Since mRNA has been exposed in a very positive way to therapeutic actions in numerous diseases, the challenges for mRNA production started to be addressed in a more intensive way, in order to solve this problematic." This sentence is long and makes little sense (did the authors mean to write 'problem' here?), and would probably be easier just to cut out entirely.

The authors reformulated the sentence referred by the Reviewer (please see page 5, lines 179-181).

  1. P8, L280: "Physical methods are simple and consist in employing ..." -- 'in' is incorrect here, 'of' should be used. ('Consists in' is repeated in P14, L512 and P26, L651 and 656; these should also be fixed).

The authors would like to apologize for the misspelling error. These structural and grammatical errors were revised throughout all the manuscript, as suggested by the Reviewer.

  1. P9, L306-7: "Different types of non-viral systems (polymeric, lipid and hybrid systems) have evolved to protect and improve the delivery of mRNA being mainly exploited the [43]." -- unfinished sentence, also 'evolved' is not correct in this context, as these are synthetic materials and not biological organisms (consider using 'developed' instead).

The authors apologize for this error. The sentence has been revised (please see page 9, lines 306-308).

  1. P10, L357-361 is a run-on sentence and should be revised.

The authors acknowledge the recommendation. The sentence has been revised and modified (see page 10, lines 357-361).

  1. P27, L315: 'Kozak' is misspelled here.

The authors would like to apologize for the misspelling error (please see page 27, line 718).

Reviewer 3 Report

I believe the manuscript is ready to be published. I have no further comments.

Author Response

Reviewer #3

I believe the manuscript is ready to be published. I have no further comments.

 Comments:

The authors acknowledge the contribution made by the Reviewer and are grateful for the careful reading.
